# Continual Auxiliary Task Learning

**Matthew McLeod, Chunlok Lo, Matthew Schlegel, Andrew Jacobsen, Raksha Kumaraswamy**
Department of Computing Science, University of Alberta
{mmcleod2,chunlok,mkschleg,ajjacobs,kumarasw}@ualberta.ca

**Martha White, Adam White**
Department of Computing Science, University of Alberta
CIFAR Canada AI Chair, Alberta Machine Intelligence Institute (Amii)
{whitem,amw8}@ualberta.ca

## Abstract

Learning auxiliary tasks, such as multiple predictions about the world, can provide many benefits to reinforcement learning systems. A variety of off-policy learning algorithms have been developed to learn such predictions, but as yet there is little work on how to adapt the behavior to gather useful data for those off-policy predictions. In this work, we investigate a reinforcement learning system designed to learn a collection of auxiliary tasks, with a behavior policy learning to take actions to improve those auxiliary predictions. We highlight the inherent non-stationarity in this continual auxiliary task learning problem, for both prediction learners and the behavior learner. We develop an algorithm based on successor features that facilitates tracking under non-stationary rewards, and prove the separation into learning successor features and rewards provides convergence rate improvements. We conduct an in-depth study into the resulting multi-prediction learning system.

## 1 Introduction

In never-ending learning systems, the agent often faces long periods of time when the external reward is uninformative. A smart agent should use this time to practice reaching subgoals, learning new skills, and refining model predictions. Later, the agent should use this prior learning to efficiently maximize external reward. The agent engages in this self-directed learning during times when the primary drives of the agent (e.g., hunger) are satisfied. Other times, the agent might have to trade-off directly acting towards internal auxiliary learning objectives and taking actions that maximize reward.

In this paper we investigate how an agent should select actions to balance the needs of several auxiliary learning objectives in a *no-reward setting* where no external reward is present. In particular, we assume the agent's auxiliary objectives are to learn a diverse set of value functions corresponding to a set of fixed policies. Our solution at a high-level is straightforward. Each auxiliary value function is learned in parallel and off-policy, and the behavior selects actions to maximize learning progress. Prior work investigated similar questions in a state-less bandit like setting, where both off-policy learning and function approximation are not required [Linke et al., 2020].

Otherwise, the majority of prior work has focused on how the agent could make use of auxiliary learning objectives, not how behavior could be used to improve auxiliary task learning. Some work has looked at defining (predictive) features, such as successor features and a basis of policies [Barreto et al., 2018, Borsa et al., 2019, Barreto et al., 2020, 2019]; universal value function approximators [Schaul et al., 2015]; and features based on value predictions [Schaul and Ring, 2013, Schlegel et al., 2021]. The other focus has been exploration, using auxiliary learning objectives to generate bonuses to aid exploration on the main task [Pathak et al., 2017, Stadie et al., 2015, Badia et al., 2020, Burda et al., 2019]; using a given set of policies in a call-return fashion for scheduled auxiliary control

[Riedmiller et al., 2018]; and discovering subgoals in environments where it is difficult for the agent to reach particular parts of the state-action space [Machado et al., 2017, Colas et al., 2019, Zhang et al., 2020, Andrychowicz et al., 2017, Pong et al., 2019]. In all of these works, the behavior was either fixed or optimized for the main task.

The problem of adapting the behavior to optimize many auxiliary predictions in the absence of external reward is sufficiently complex to merit study in isolation. It involves several inter-dependent learning mechanisms, multiple sources of non-stationarity, and high-variance due to off-policy updating. If we cannot design learning systems that efficiently learn their auxiliary objectives in isolation, then the agent is unlikely to learn its auxiliary tasks while additionally balancing external reward maximization.

Further, understanding how to efficiently learn a collection of auxiliary objectives is complementary to the goals of using those auxiliary objectives. It could amplify the auxiliary task effect in UNREAL [Jaderberg et al., 2017], improve the efficiency and accuracy of learning successor features and universal value function approximators, and improve the quality of the sub-policies used in scheduled auxiliary control. It can also benefit the numerous systems that discover options, skills, and subgoals [Gregor et al., 2017, Eysenbach et al., 2019a, Veeriah et al., 2019, Pitis et al., 2020, Nair et al., 2020, Pertsch et al., 2020, Colas et al., 2019, Eysenbach et al., 2019b], by providing improved algorithms to learn the resulting auxiliary tasks. For example, for multiple discovered subgoals, the agent can adapt its behavior to efficiently learn policies to reach each subgoal.

In this paper we introduce an architecture for parallel auxiliary task learning. As the first such work to tackle this question in reinforcement learning with function approximation, numerous algorithmic challenges arise. We first formalize the problem of learning multiple predictions as a reinforcement learning problem, and highlight that the rewards for the behavior policy are inherently non-stationary due to changes in learning progress over time. We develop a strategy to use successor features to exploit the stationarity of the dynamics, whilst allowing for fast tracking of changes in the rewards, and prove that this separation provides a faster convergence rate than standard value function algorithms like temporal difference learning. We empirically show that this separation facilitates tracking both for prediction learners with non-stationary targets as well as the behavior.

## 2    Problem Formulation

We consider the *multi-prediction problem*, in which an agent continually interacts with an environment to obtain accurate predictions. This interaction is formalized as a Markov decision process (MDP), defined by a set of states $\mathcal{S}$, a set of actions $\mathcal{A}$, and a transition probability function $\mathcal{P}(s, a, s')$. The agent's goal, when taking actions, is to gather data that is useful for learning $N$ predictions, where each prediction corresponds to a general value function (GVF) [Sutton et al., 2011].

A GVF question is formalized as a three tuple $(\pi, \gamma, c)$, where the target is the expected return of the cumulant, defined by $c : \mathcal{S} \times \mathcal{A} \times \mathcal{S} \to \mathbb{R}$, when following policy $\pi : \mathcal{S} \times \mathcal{A} \to [0, 1]$, discounted by $\gamma : \mathcal{S} \times \mathcal{A} \times \mathcal{S} \to [0, 1]$. More precisely, the target is the action-value

$$Q(s, a) \stackrel{\text{def}}{=} \mathbb{E}_\pi \left[ G_t | S_t = s, A_t = a \right] \quad \text{for } G_t \stackrel{\text{def}}{=} C_{t+1} + \gamma_{t+1} G_{t+1}$$

where $C_{t+1} \stackrel{\text{def}}{=} c(S_t, A_t, S_{t+1})$ and $\gamma_{t+1} \stackrel{\text{def}}{=} \gamma(S_t, A_t, S_{t+1})$. The extension of $\gamma$ to transitions allows for a broader class of problems, including easily specifying termination, without complicating the theory [White, 2017]. The expectation is under policy $\pi$, with transitions according to $\mathcal{P}$. The prediction targets could also be state-value functions; we assume the targets are action-values in this work to provide a unified discussion of successor features for both the GVF and behavior learners.

At each time step, the agent produces $N$ predictions, a $\hat{Q}_t^{(j)}(S_t, A_t)$ for prediction $j$ with true targets $Q_t^{(j)}(S_t, A_t)$. We assume the GVF question can change over time, and so $Q$ can change with time. The goal is to have low error in the prediction, in terms of the root mean-squared value error (RMSVE), under state-action weighting $d : \mathcal{S} \times \mathcal{A} \to \mathbb{R}$:

$$\text{RMSVE}(\hat{Q}, Q) \stackrel{\text{def}}{=} \sqrt{\sum_{s \in \mathcal{S}} \sum_{a \in \mathcal{A}} d(s, a)(\hat{Q}(s, a) - Q(s, a))^2} \tag{1}$$

The total error up to time step $t$, across all predictions, is TE $\stackrel{\text{def}}{=} \sum_{i=1}^{t} \sum_{j=1}^{N} \text{RMSVE}(\hat{Q}_i^{(j)}, Q_i^{(j)})$.

The agent's goal is to gather data and update its predictions to make TE small. This goal can itself be formalized as a reinforcement learning problem, by defining rewards for the behavior policy that depend on the agent's predictions. Such rewards are often called *intrinsic rewards*. For example, if we could directly measure the RMSVE, one potential intrinsic reward would be the decrease in the RMSVE after taking action $A_t$ from state $S_t$ and transitioning to $S_{t+1}$. This reflects the agent's learning progress—how much it was able to learn—due to that new experience. The reward is high if the action generated data that resulted in substantial learning. While the RMSVE is the most direct measure of learning progress, it cannot be calculated without the true values.

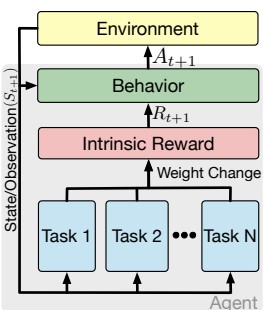

Many intrinsic rewards have been considered to estimate the learning progress of predictions. A recent work provided a thorough survey of different options, as well as an empirical study [Linke et al., 2020]. Their conclusion was that, for reasonable prediction learners, simple learning progress measures—like the change in weights—were effective for producing effective data gathering. We rely on this conclusion here, and formalize the problem using the $\ell_1$ norm on the change in weights. Other intrinsic rewards could be swapped into the framework; but, because our focus is on the non-stationarity in the system and because empirically we found this weight-change intrinsic reward to be effective, we opt for this simple choice upfront.

We provide the generic pseudocode for a multi-prediction reinforcement learning system, in Algorithm 1. Note that the behavior agent also has a separate transition-based $\gamma$, which enables us to encode

---

**Algorithm 1** Multi-Prediction Learning System
**Input:** $N$ GVF questions
  Initialize behavior policy parameters $\theta_0$
  and GVF learners $w_0^{(1)}, \ldots, w_0^{(N)}$
  Obtain initial observation $S_0$
  **for** $t = 0, 1, \ldots$ **do**
    Choose action $A_t$ according to $\pi_{\theta_t}(\cdot|S_t)$
    Observe next state vector $S_{t+1}$ and $\gamma_{t+1}$
    // Update predictions with new data
    **for** $j = 1$ **to** $N$ **do**
      $c \leftarrow c^{(j)}(S_t, A_t, S_{t+1})$
      $\gamma \leftarrow \gamma^{(j)}(S_t, A_t, S_{t+1})$
      Update $w_t^{(j)}$ with $(S_t, A_t, c, S_{t+1}, \gamma)$
    // Compute intrinsic reward, update behavior
    $R_{t+1} \leftarrow \sum_{j=1}^{N} \|w_{t+1}^{(j)} - w_t^{(j)}\|_1$
    Update $\theta_t$ with $(S_t, A_t, R_{t+1}, S_{t+1}, \gamma_{t+1})$

---

both continuing and episodic problems. For example, the pseudo-termination for a GVF could be a particular state in the environment, such as a doorway. The discount for the GVF would be zero in that state, even though it is not a true terminal state; the behavior discount $\gamma_{t+1}$ would not be zero.

## 3 Non-stationarity Induced by Learning

On the surface, the multi-prediction problem outlined in the previous section is a relatively straightforward reinforcement learning problem. The behavior policy learns to maximize cumulative reward, and simultaneously learns predictions about its environment. Many RL systems incorporate prediction learning, either as auxiliary tasks or to learn a model. However, unlike standard RL problems, the rewards for the behavior are non-stationary when using intrinsic rewards, even under stationary dynamics. Further, the prediction problems themselves are non-stationary due to a changing behavior.

To understand this more deeply, consider first the behavior rewards. On each time step, the predictions are updated. Progressively, they get more and more accurate. Imagine a scenario where they can become perfectly accurate, such as in the tabular setting with stationary cumulants. The behavior rewards are high in early learning, when predictions are inaccurate. As predictions become more and more accurate, the change in weights gets smaller until eventually the behavior rewards are near zero. This means that when the behavior revisits a state, the reward distribution has actually changed. More generally, in the function approximation setting, the behavior rewards will continue to change with time, not necessarily decay to zero.

The prediction problems are also non-stationary for two reasons. First, the cumulants themselves might be non-stationary, even if the transition dynamics are stationary. For example, the cumulant could correspond to the amount of food in a location in the environment, that slowly gets depleted. Or, the cumulant could depend on a hidden variable, that makes the outcome appear non-stationary. Even with a stationary cumulant, the prediction learning problem can be non-stationary due to a changing behavior policy. As the behavior policy changes, the state distribution changes. Implicitly,

when learning off-policy, the predictions are minimizing an objective weighted by the state visitation under the behavior policy. As the behavior changes, the underlying objective is actually changing, resulting in a non-stationary prediction problem.

Though there has been some work on learning under non-stationarity in RL and bandits, none to our knowledge has addressed the multi-prediction setting in MDPs. There has been some work developing reinforcement learning algorithms for non-stationary MDPs, but largely for the tabular setting [Sutton and Barto, 2018, Da Silva et al., 2006, Abdallah and Kaisers, 2016, Cheung et al., 2020] or assuming periodic shifts [Chandak et al., 2020a,b, Padakandla et al., 2020]. There has also been some work in the non-stationary multi-armed bandit setting [Garivier and Moulines, 2008, Koulouriotis and Xanthopoulos, 2008, Besbes et al., 2014]. The non-stationary rewards for the behavior, that decay over time, have been considered for the bandit setting, under rotting bandits [Levine et al., 2017, Seznec et al., 2019]; these algorithms do not obviously extend to the RL setting.

# 4 Handling the Non-Stationarity in a Multi-prediction System

In this section, we describe a unified approach to handle non-stationarity in both the GVF and behavior learners, using successor features. We first discuss how to use successor features to learn under non-stationary cumulants, for prediction. Then we discuss using successor features for control, allows us to leverage this approach for non-stationary rewards for the behavior. We then discuss state-reweightings, and how to mitigate non-stationarity due to a changing behavior.

## 4.1 Successor Features for Non-stationary Rewards

Successor features provide an elegant way to learn value functions under non-stationarity. The separation of learning stationary successor features and rewards enables more effective tracking of non-stationary rewards, as we explain in this section and formally prove in Section 5.

Assume that there is a weight vector $\mathbf{w}^* \in \mathbb{R}^d$ and features $\mathbf{x}(s,a) \in \mathbb{R}^d$ for each state and action $(s,a)$ such that $r(s,a) = \langle \mathbf{x}(s,a), \mathbf{w}^* \rangle$. Recursively define

$$\boldsymbol{\psi}(s,a) = \mathbb{E}_\pi[\mathbf{x}(S_t, A_t) + \gamma_{t+1} \boldsymbol{\psi}(S_{t+1}, A_{t+1})|S_t = s, A_t = a]$$

$\boldsymbol{\psi}(s,a)$ is called the *successor features*, the discounted cumulative sum of feature vectors, if we follow policy $\pi$. For $\boldsymbol{\psi}_t \stackrel{\text{def}}{=} \boldsymbol{\psi}(S_t, A_t)$ and $\mathbf{x}_t \stackrel{\text{def}}{=} \mathbf{x}(S_t, A_t)$, we can see $Q(s,a) = \langle \boldsymbol{\psi}(s,a), \mathbf{w}^* \rangle$

$$
\begin{aligned}
\langle \boldsymbol{\psi}(s,a), \mathbf{w}^* \rangle &= \mathbb{E}_\pi[\langle \mathbf{x}_t, \mathbf{w}^* \rangle|S_t = s, A_t = a] + \mathbb{E}_\pi[\gamma_{t+1}\langle \boldsymbol{\psi}_{t+1}, \mathbf{w}^* \rangle|S_t = s, A_t = a] \\
&= r(s,a) + \mathbb{E}_\pi[\gamma_{t+1}\langle \mathbf{x}_{t+1}, \mathbf{w}^* \rangle|S_t = s, A_t = a] + \mathbb{E}_\pi[\gamma_{t+1}\gamma_{t+2}\langle \boldsymbol{\psi}_{t+2}, \mathbf{w}^* \rangle|S_t = s, A_t = a] \\
&= r(s,a) + \mathbb{E}_\pi[\gamma_{t+1}r_{t+1}|S_t = s, A_t = a] + \mathbb{E}_\pi[\gamma_{t+1}\gamma_{t+2}\langle \boldsymbol{\psi}_{t+2}, \mathbf{w}^* \rangle|S_t = s, A_t = a] \\
&= \quad \ldots \quad = \mathbb{E}_\pi[r(s,a) + \gamma_{t+1}r_{t+1} + \gamma_{t+1}\gamma_{t+2}r_{t+2} + \ldots |S_t = s, A_t = a] \quad = Q(s,a).
\end{aligned}
$$

If we have features $\mathbf{x}(s,a) \in \mathbb{R}^d$ which allow us to represent the immediate reward, then successor features provide a good representation to approximate the GVF. We simply learn another set of parameters $\mathbf{w}_c \in \mathbb{R}^d$ that predict the immediate cumulant (or reward): $c(s,a) \approx \langle \mathbf{x}(s,a), \mathbf{w}_c \rangle$. These parameters $\mathbf{w}_c$ are updated using a standard regression update, and $Q(s,a) \approx \langle \boldsymbol{\psi}(s,a), \mathbf{w}_c \rangle$.

The successor features $\boldsymbol{\psi}(s,a)$ themselves, however, also need to be approximated. In most cases, we cannot explicitly maintain a separate $\boldsymbol{\psi}(s,a)$ for each $(s,a)$, outside of the tabular setting. Notice that each element in $\boldsymbol{\psi}(s,a)$ corresponds to a true expected return: the cumulative discounted sum of a reward feature into the future. Therefore, $\boldsymbol{\psi}(s,a)$ can be approximated using any value function approximation method, such as temporal difference (TD) learning. We learn parameters $\mathbf{w}_\psi$ for the approximation $\hat{\boldsymbol{\psi}}(s,a;\mathbf{w}_\psi) = [\hat{\psi}_1(s,a;\mathbf{w}_\psi), ..., \hat{\psi}_d(s,a;\mathbf{w}_\psi)]^\top \in \mathbb{R}^d$ where $\hat{\psi}_m(s,a;\mathbf{w}_\psi) \approx \psi_m(s,a)$. We can use any function approximator for $\hat{\boldsymbol{\psi}}(s,a;\mathbf{w}_\psi)$, such as linear function approximation with tile coding with $\mathbf{w}_\psi$, linearly weighting the tile coding features to produce $\hat{\boldsymbol{\psi}}(s,a;\mathbf{w}_\psi)$, or neural networks, where $\mathbf{w}_\psi$ are the parameters of the neural network.

We summarize the algorithm using successor features for non-stationary rewards/cumulants, called SF-NR, in Algorithm 2. We provide an update formula for the approximate SF using Expected Sarsa for prediction [Sutton and Barto, 2018] for simplicity, but note that any value learning algorithm can be used here. In our experiments, we use Tree-Backup [Precup, 2000] because it reduces variance

from off-policy learning; we provide the pseudocode in Appendix D. Algorithm 2 assumes that the reward features $\mathbf{x}(s,a)$ are given, but of course these can be learned as well. Ideally, we would learn a compact set of reward features that provide accurate estimates as a linear function of these reward features. A compact (smaller) set of reward features is preferred because it makes the SF more computationally efficient to learn.

There are two key advantages from the separation into learning successor features and immediate cumulant estimates. First, it easily allows different or changing cumulants to be used, for the same policy, using the same successor features. The transition dynamics summarized in the stationary successor features can be learned slowly to high accuracy and re-used. This re-use property is why these representations have been used for transfer [Barreto et al., 2017, 2018, 2020]. This property is pertinent for us, because it allows us to more easily track changes in the cumulant. The regression updates can quickly update the parameters $\mathbf{w}_c$, and exploit the already learned successor features to more quickly track value estimates. Small changes in the rewards can result in large changes in the values; without the separation, therefore, it can be more difficult to directly track the value estimates.

---

**Algorithm 2** Successor Features for Non-stationary Rewards (SF-NR)

---

**Input:** $(S_t, A_t, S_{t+1}, C_{t+1}, \gamma_{t+1}), \pi, \mathbf{w}_\psi, \mathbf{w}_c$

$\quad \mathbf{x} \leftarrow \mathbf{x}(S_t, A_t)$

$\quad \hat{\psi} \leftarrow \hat{\psi}(S_t, A_t; \mathbf{w}_\psi)$

$\quad \hat{\psi}' \leftarrow \sum_{a'} \pi(a'|S_{t+1}) \hat{\psi}(S_{t+1}, a'; \mathbf{w}_\psi)$

$\quad \Delta \leftarrow \mathbf{0}$

$\quad$ **for** $m = 1$ **to** $d$ **do**

$\quad\quad \delta_m \leftarrow \mathbf{x}_m + \gamma_{t+1}\hat{\psi}'_m - \hat{\psi}_m$

$\quad\quad \Delta \leftarrow \Delta + \delta_m \nabla \hat{\psi}_m$

$\quad \mathbf{w}_\psi \leftarrow \mathbf{w}_\psi + \alpha\Delta$

$\quad \mathbf{w}_c \leftarrow \mathbf{w}_c + \alpha(C_{t+1} - \langle \mathbf{x}, \mathbf{w}_c \rangle)\mathbf{x}$

---

Second, the separation allows us to take advantage of online regression algorithms with strong convergence guarantees. Many optimizers and accelerations are designed for a supervised setting, rather than for temporal difference algorithms. Once the successor features are learned, the prediction problem reduces to a supervised learning problem. We can therefore even further improve tracking by leveraging these algorithms to learn and track the immediate cumulant. We formalize the convergence rate improvements, from this separation, in Section 5.

### 4.2 GPI with Successor Features for Control

In this section we outline a control algorithm under non-stationary rewards. SF-NR provides a method for updating the value estimate due to changing rewards. The behavior for the multi-prediction problem has changing rewards, and so could benefit from SF-NR. But SF-NR only provides a mechanism to efficiently track action-values for a fixed policy, not for a changing policy. Instead, we turn to the idea of constraining the behavior to act greedily with respect to the values for a set of policies, introduced as Generalized Policy Improvement (GPI) Barreto et al. [2018, 2020].

For our system, this is particularly natural, as we are already learning successor features for a collection of policies. Let us start there, where we assume our set of policies is $\Pi = \{\pi_1, \ldots, \pi_N\}$. Assume also that we have learned the successor features for these policies, $\hat{\psi}(s, a; \mathbf{w}_\psi^{(j)})$, and that we have weights $\theta_r \in \mathbb{R}^d$ such that $\langle \mathbf{x}(s,a), \theta_r \rangle \approx \mathbb{E}[R_{t+1}|S_t = s, A_t = a]$ for behavior reward $R_{t+1}$. Then on each step, the behavior policy takes the following greedy action

$$\mu(s) = \operatorname*{argmax}_a \max_{j \in \{1, \ldots, N\}} \hat{Q}_r^{(j)}(s, a) = \operatorname*{argmax}_a \max_{j \in \{1, \ldots, N\}} \langle \hat{\psi}(s, a; \mathbf{w}_\psi^{(j)}), \theta_r \rangle$$

The resulting policy is guaranteed to be an improvement: in every state the new policy has a value at least as good as any of the policies in the set [Barreto et al., 2017, Theorem 1]. Later work also showed sampled efficiency of GPI when combining known reward weights to solve novel tasks [Barreto et al., 2020].

The use of successor features has similar benefits as discussed above, because the estimates can adapt more rapidly as the rewards change, due to learning progress changing over time. The separation is even more critical here, as we know the rewards are constantly drifting, and tracking quickly is even more critical. We could even more aggressively adapt to these non-stationary rewards, by anticipating trends. For example, instead of a regression update, we can model the trend (up or down) in the reward for a state and action. If the reward has been decreasing over time, then likely it will continue to decrease. Stochastic gradient descent will put more weight on recent points, but would likely predict a higher expected reward than is actually observed. For simplicity here, we still choose to use stochastic gradient descent, as it is a reasonably effective tracking algorithm, but note that performance improvements could likely be obtained by exploiting this structure in this problem.

We can consider a different set of policies for GVFs and behavior. However, the two are naturally coupled. First, the GPI theory shows that greedifying over a larger collection of policies provides better policies. It is sensible then to at least include the GVF policies into the set for the behavior. Second, the behavior needs to learn the successor features for the additional policies. Arguably, it should try to gather data to learn these well, so as to facilitate its own policy improvement. It should therefore also incorporate the learning progress for these successor features, into the intrinsic reward. For this work, therefore, we assume that the behavior uses the set of GVF policies. Note that the weight change intrinsic reward uses the concatenation of $\mathbf{w}_\psi$ and $\mathbf{w}_c$.

### 4.3 Interest and prior corrections for the changing state distribution

The final source of non-stationarity is in the state distribution. As the behavior $\mu$ changes, the state-action visitation distribution $d_{\mu_t} : \mathcal{S} \times \mathcal{A} \to [0, 1]$ changes. The state distribution implicitly weights the relative importance of states in the GVF objective, called the projected Bellman error (PBE). Correspondingly, the optimal SF solution could be changing, since the objective is changing. The impact of a changing state-weighting depends on function approximation capacity, because the weighting indicates how to trade-off function approximation error across states. When approximation error is low or zero—such as in the tabular setting—the weighting has little impact on the solution. Generally, however, we expect some approximation error and so a non-negligible impact.

We can completely remove this source of non-stationary by using *prior corrections*. These are products of importance sampling ratios, that reweight the trajectory to match the probability of seeing that trajectory under the target policy $\pi$. Namely, it modifies the state weighting to $d_\pi$, the state-action visitation distribution under $\pi$. We explicitly show this in Appendix C. Unfortunately, prior corrections can be highly problematic in a system where the behavior policy takes exploratory actions and target policies are nearly deterministic. It is likely that these corrections will often either be zero, or near zero, resulting in almost no learning.

To overcome this inherent difficulty, we restrict which states are important for each predictive question. Likely, when creating a GVF, the agent is interested in predictions for that GVF only in certain parts of the space. This is similar to the idea of initiation sets for options, where an option is only executed from a small set of relevant states. We can ask: what is the GVF answer, from this smaller set of states of interest? This can be encoded with a non-negative interest function, $i(s, a)$, where some (or even many) states have an interest of zero. This interest is incorporated into the state-weighting in the objective, so the agent can focus function approximation resources on states of interest.

When using interest, it is sensible to use emphatic weights [Sutton et al., 2016]. Emphatic weightings are a prior correction method, used under the excursions model [Patterson et al., 2021]. They reweight to a discounted state-action visitation under $\pi$ when starting from states proportionally to $d_\mu$. Further, they ensure states inherit the interest of any states that bootstrap off of them. Even if a state has an interest of zero, we want to accurately estimate its value if an important states bootstraps off of its value. The combination of interest and emphatic weightings—which shift state-action weighting to visitation under $\pi$—means that we mitigate much of the non-stationarity in the state-action weighting. We provide the pseudocode for this Emphatic TB (ETB) algorithm in Appendix D.

## 5 Sample Efficiency of SF-NR

As suggested in Section 4.1, the use of successor features makes SF-NR particularly well-suited to our multi-prediction problem setting. The reason for this is simple: given access to an accurate SF matrix, value function estimation reduces to a *fundamentally simpler* linear prediction problem. Indeed, access to an accurate SF enables one to sidestep known lower-bounds on PBE estimation.

For simplicity, we prove the result for value functions; the result easily extends to action-values. Denote by $\mathbf{v}^\pi \in \mathbb{R}^{|\mathcal{S}|}$ the vector with entries $v^\pi(s)$, $\mathbf{r}^\pi \in \mathbb{R}^{|\mathcal{S}|}$ the vector of expected immediate rewards in each state, and $\mathbf{P} \in \mathbb{R}^{|\mathcal{S}| \times |\mathcal{S}|}$ the matrix of transition probabilities. The following lemma, proven in Appendix A.1, relates mean squared value error (VE) to one-step reward prediction error.

**Lemma 1** *Assume there exists a $\mathbf{w}^* \in \mathbb{R}^d$ such that $\mathbf{r}^\pi = \mathbf{X}\mathbf{w}^*$. Let $\hat{\mathbf{r}} \overset{def}{=} \mathbf{X}\mathbf{w}$ for some $\mathbf{w} \in \mathbb{R}^d$, and let $\mathbf{D} = Diag(\{d(s)\}_{s \in \mathcal{S}})$ for distribution $d$ fully supported on $\mathcal{S}$, with $\|\cdot\|_\mathbf{D}$ the weighted norm under $\mathbf{D}$. Then the value estimate $\hat{\mathbf{v}} \overset{def}{=} \Psi\mathbf{w}$ satisfies $\frac{1}{2}\|\mathbf{v}^\pi - \hat{\mathbf{v}}\|_\mathbf{D}^2 \leq \frac{\|\mathbf{r}^\pi - \hat{\mathbf{r}}\|_\mathbf{D}^2}{2(1-\gamma)^2}$.*

Thus we can ensure that $\mathrm{VE}(\mathbf{v}^\pi, \hat{\mathbf{v}}) \le \varepsilon$ by ensuring that $\|\mathbf{r}^\pi - \hat{\mathbf{r}}\|_{\mathbf{D}}^2 \le \varepsilon(1-\gamma)^2$. This is promising, as this latter expression is readily expressed as the objective of a linear regression problem. To illustrate the utility of this, let's look at a concrete example: suppose the agent has an accurate SF matrix $\Psi$ and that the reward function changes at some point in the agent's deployment. Suppose access to a batch of transitions $\mathcal{D} \stackrel{\text{def}}{=} \{S_t, A_t, S_t', r_t, \rho_t\}_{t=1}^T$ with which we can correct our estimate of $v^\pi$, where each $(s, a, s', \rho) \in \mathcal{D}$ is such that $s \sim d_\mu$ for some known behavior policy $\mu$, $A_t \sim \pi(\cdot|s)$, $s' \sim P(\cdot|s, a)$ and $r = r(s, a, s')$. Assume for simplicity that $\rho_t \le \rho_{\max}$, $\|\phi(S_t)\|_\infty \le L$, $r_t \le R_{\max}$ for some finite $\rho_{\max}, R_{\max}, L \in \mathbb{R}_+$. Then we can get the following result, proven in Appendix A.2, that is a straightforward application of Orabona [2019, Theorem 7.26].

**Proposition 1** *Define* $\ell_t(w) \stackrel{\text{def}}{=} \frac{\rho_t}{2}(r_t - \langle x(S_t), w \rangle)^2$. *Suppose we apply a basic recursive least-squares estimator to minimize regret on this loss sequence, producing a sequence of iterates* $w_t$. *Let* $\overline{w}_T \stackrel{\text{def}}{=} \frac{1}{T}\sum_{t=1}^T w_t$ *denote the average iterate. For* $\hat{v}(s) = \langle \psi(s), \overline{w}_T \rangle$, *we have that*

$$\|\mathbf{v}^\pi - \hat{\mathbf{v}}\|_{\mathbf{D}}^2 \le O\left(\frac{d\rho_{\max}R_{\max}^2 \log\left(1 + \rho_{\max}L^2 T\right)}{(1-\gamma)^2 T}\right). \tag{2}$$

In contrast, without the SF we are faced with minimizing a harder objective: the PBE. It can be shown that minimizing the PBE is equivalent to a stochastic saddle-point problem, and the convergence to the saddle-point of this problem has an unimprovable rate of $O\left(\frac{\tau}{T^2} + \frac{(1+\gamma)\rho_{\max}L^2 d}{T} + \frac{\sigma}{\sqrt{T}}\right)$ where $\tau$ is the maximum eigenvalue of the covariance matrix and $\sigma$ bounds gradient stochasticity, and this convergence rate translates into the performance bound $\frac{1}{2}\|\mathbf{v}^\pi - \hat{\mathbf{v}}\|_{\mathbf{D}}^2 \le O\left(\sqrt{\frac{\tau}{T^2} + \frac{(1+\gamma)\rho_{\max}L^2 d}{T} + \frac{\sigma}{\sqrt{T}}}\right)$ [Liu et al., 2018a, Proposition 5]. Comparing with Equation 2, we observe an additional dependence of $O(\sqrt{\tau}/T)$ as well as the worse dependence of at least $O(1/\sqrt{T}) \ge (\log(T)/T)$ on all other quantities of interest, reinforcing the intuition that access to the SF enables us to more efficiently re-evaluate the value function.

# 6 A First Experiment Testing the Multi-prediction System

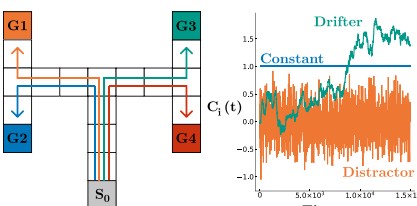

Figure 1: **Tabular TMaze** with 4 GVFs, with cumulants of zero except in the goals. The right plot shows the cumulants in the goals. G2 and G4 have constant cumulants, G1 has a distractor cumulant and G4 a drifter.

In this section, we investigate the utility of using SF-NR under non-stationary cumulants and rewards, both for prediction and control. We conduct the experiment in a TMaze environment, inspired by the environments used to test animal cognition [Tolman and Honzik, 1930]. The environment, depicted in Figure 1, has four GVFs where each policy takes the fastest route to its corresponding goal. The cumulants are zero everywhere except for at the goals. The cumulant can be of three types: a constant fixed value (**constant**), a fixed-mean and high variance value (**distractor**), or a non-stationary zero-mean random walk process with a low variance (**drifter**). Exact formulas for these cumulants are in Appendix E.1.

**Utility of SF-NR for a Fixed Behavior Policy**

We start by testing the utility of SF-NR for GVF learning, under a fixed policy that provides good data coverage for every GVF. The *Fixed-Behavior Policy* is started from random states in the TMaze, and moves towards the closest goal, with a 50/50 chance of going either direction if there is a tie. This policy is like a round robin policy, in that one of the GVF policies is executed each episode and, in expectation, all four policies are executed the same number of times.

We compare an agent that uses SF-NR and one that learns the approximate GVFs using Tree Back-Up (TB). TB is an off-policy temporal difference (TD) algorithm, that reduces variance in the eligibility trace. We also use TB to learn the successor features in SF-NR. Both use $\lambda = 0.9$ and a stepsize method called Auto [Mahmood et al., 2012] designed for online learning. We sweep the initial stepsize and meta stepsizes for Auto. For further details about the agents and optimizer, see Appendix D. We additionally compare to least squares TD (LSTD), with $\lambda = 0.9$, particularly as it computes a matrix similar to the SF, but does not separate out cumulant learning (see Appendix B for this connection).

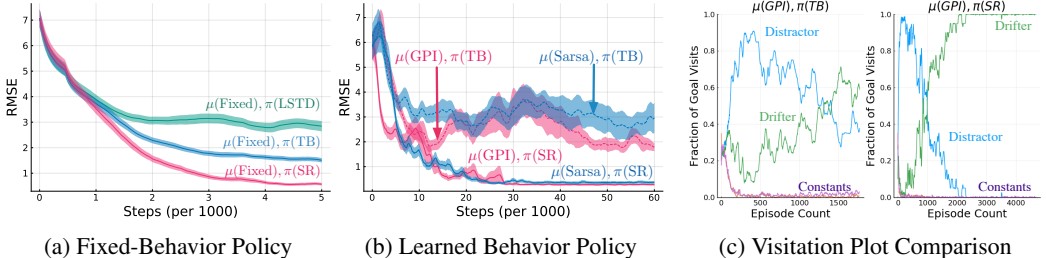

| (a) Fixed-Behavior Policy | (b) Learned Behavior Policy | (c) Visitation Plot Comparison |

Figure 2: Performance in **Tabular TMaze**, with averages over 30 runs. **(a)** and **(b)** show average off-policy prediction RMSE, with standard errors, where the error is weighted by (a) the state distribution $d_\mu$ for the Fixed-Behavior policy and (b) a uniform state weighting when learning the behavior. **(c)** Goal visitation plots for GPI with SF and TB.

In Figure 2a, we can see SF-NR allows for much more effective learning, particularly later in learning when it more effectively tracks the non-stationary signals. LSTD performs much more poorly, likely because it corresponds to a closed-form batch solution, which uses old cumulants that are no longer reflective of the current cumulant distribution.

**Investigating GPI for Learning the Behavior**

Next we investigate if SF-NR improves learning of the whole system, both for the GVFs and for the behavior policy. We use SF-NR and TB for the GVF learners, and Expected Sarsa (Sarsa) and GPI for the behavior. The GPI agent uses the GVF policies for its set of policies. The reward features for the behavior are likely different than those for the GVF learners, because the cumulants are zero in most states whereas intrinsic rewards are likely non-zero in most states. The GPI agent, therefore, learns its own SFs for each policy, also using TB. The reward weights that estimate the (changing) intrinsic rewards are learned using Auto, as are the SFs. Note that the behavior and GVF learners all share the same meta-step size—namely only one shared parameter is swept.

The results highlight that SF for the GVFs is critical for effective learning, though GPI and Sarsa perform similarly, as shown in Figure 3b. The utility of SF is even greater here, with TB GVF learners inducing much worse performance than SF GVF learners. GPI and Sarsa are similar, which is likely due to the fact that Sarsa uses traces with tabular features, which allow states along the trajectory to the drifter goal to update quickly. In following sections, we find a bigger distinction between the two.

We visualize the goal visitation of GPI in Figure 2c. Once the GVF learners have a good estimate for the *constant* cumulant signals and the *distractor* cumulant signal, the agent behavior should switch to visiting only the *drifter* cumulant as that is the only goal where visiting would improve the GVF prediction. When using SF GVF learners, this behavior emerges, but under TB GVF learners the agent incorrectly focuses on the distractor. This is even more pronounced for Sarsa (see Appendix F.2).

## 7  Experiments under Function Approximation

We evaluate our system in a similar fashion to the last section, but now under function approximation. We use a benchmark problem at the end of this section, but start with experiments in the TMaze modified to be a continuous environment, with full details described in Appendix E.1. The environment observation $o_t \in \mathbb{R}^2$ corresponds to the xy coordinates of the agent. We use tile coded features of 2 tilings of 8 tiles for the state representation, both for TB and to learn the SF.

The reward features for the GVF learners can be much simpler than the state-action features because they only need to estimate the cumulants, which are zero in every state except the goals. The reward features are a one-hot encoding indicating if $s'$ in the tuple of $(s, a, s')$ is in the pseudo-termination goals of the GVFs. For the Continuous TMaze, this gives a 4 dimensional vector. The reward features for GPI is state aggregation applied along the one dimensional line components. Appendix E.1 contains more details on the reward features for the GVF and behavior learners.

**Results for a Fixed Behavior and Learned Behaviors**

Under function approximation, SF-NR continues to enable more effective tracking of the cumulants than the other methods. For control, GPI is notably better than Sarsa, potentially because under

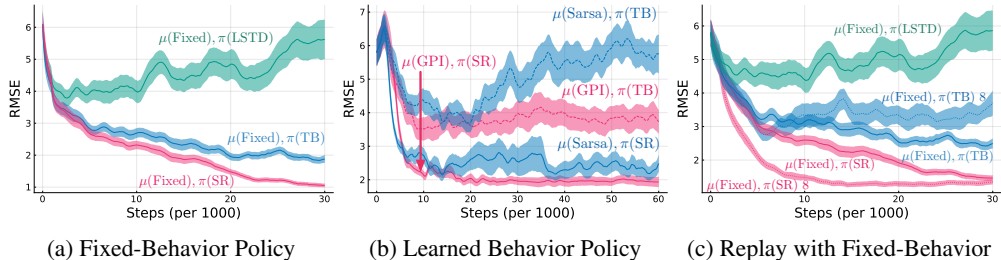

(a) Fixed-Behavior Policy     (b) Learned Behavior Policy     (c) Replay with Fixed-Behavior

Figure 3: Performance in **Continuous TMaze**, with averages over 30 runs. **(a)** and **(b)** show average off-policy prediction RMSE, with standard errors, where the error is weighted by (a) the state distribution $d_\mu$ for the Fixed-Behavior policy and (b) a uniform state weighting when learning the behavior. **(c)** RMSE in Continuous TMaze with a Fixed Behavior when incorporating replay.

function approximation eligibility traces are not as effective at sweeping back changes in behavior rewards and so the separation is more important. We include visitations plots in Appendix F.1, which are similar to the tabular setting.

Note that the efficacy of SF-NR and GPI relied on having reward features that did not overly generalize. The SF learns the expected feature vector when following the target policy. For the GVF learners, if states on the trajectory share features with states near the goal, then the value estimates will likely be higher for those states. The rewards are learned using squared error, which unlike other losses, is likely only to bring cumulant estimates to near zero. These small non-zero cumulant estimates are accumulated by the SF for the entire trajectory, resulting in higher error than TB. We demonstrate this in Appendix F.3. We designed reward features to avoid this problem for our experiments, knowing that effective reward features can and have been learned for SF [Barreto et al., 2020].

**Results using Replay**

The above results uses completely online learning, with eligibility traces. A natural question is if the more modern approach of using replay could significantly change the results. In fact, early versions of the system included replay but had surprisingly negative results, which we later realized was due to the inherent non-stationarity in the system. Replaying old cumulants and rewards, that have become outdated, actually harms performance of the system. Once we have the separation with the SF, however, we can actually benefit from replay for this stationary component.

We demonstrate this result in Figure 3c. We use $\lambda = 0$ for this result, because we use replay. The settings are otherwise the same as above, and we resweep hyperparameters for this experiment. SF-NR benefits from replay, because it only uses it for its stationary component: the SF. TB, on the other hand, actually performs more poorly with replay. As before, LSTD which similarly uses old cumulants, also performs poorly.

**Incorporating Interest**

To study the effects of interest, a more open world environment is needed. The *Open 2D World* is used to analyze this problem as described in Appendix E.2. At the start of each episode, the agent begins in the center of the environment. The interest for each GVF in the states is one if the state is in the same quadrant as the GVF's respective goal, and zero otherwise. This enables the GVFs to focus their learning on a subset of the entire space and thus use the function approximation resources more wisely and give a better weight change profile as an intrinsic reward to the behavior learner. Each GVF prediction $i$ is evaluated under state-action weighting induced by running $\pi_i$, with results in Figure 4.

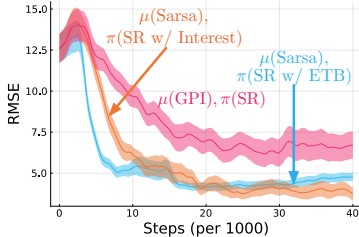

Figure 4: Using interest: shading is standard error over 30 runs.

Both TB with interest and ETB reweight states to focus more on state visitation under the policy. Both significantly improve performance over not using interest, both allowing faster learning and reaching a lower error. The reweighting under ETB more closely matches state visitation under the policy, and accounts for the impacts of bootstrapping. We find that ETB does provide some initial learning benefits. The original ETD algorithm is known to suffer from variance issues; we may find with variance reduction that the utility of ETB is even more pronounced.

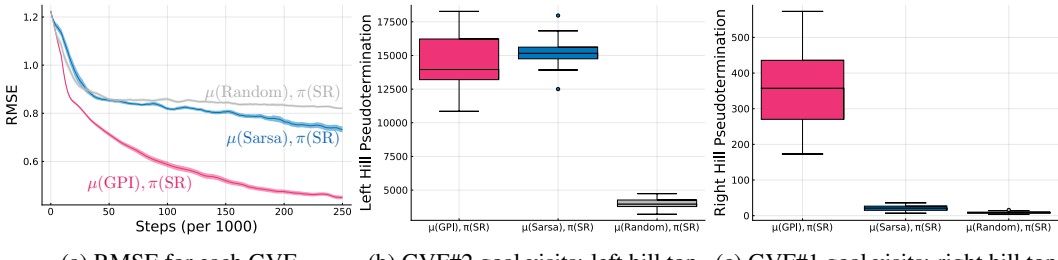

(a) RMSE for each GVF   (b) GVF#2 goal visits: left hill top   (c) GVF#1 goal visits: right hill top

Figure 5: Performance in **Mountain Car** averaged over 30 runs, with standard errors. **(a)** Learning curves for RMSE, with a uniform weighting over states and actions. **(b), (c)** show the number of times that the agent reached the termination for each GVF.

**Validation of the Multi-Prediction System in a Standard Benchmark Problem**

Finally, we investigate multi-prediction learning in an environment not obviously designed for this setting: Mountain Car. The goal here is to show that multi-prediction learning is natural in many problem settings, and to show results in a standard benchmark not designed for our setting that has more complex transition dynamics. In the usual formulation the agent must learn to rock back and forth building up momentum to reach the top of the hill on the right—a classic cost to goal problem. This is a hard exploration task where a random agent requires thousands of steps to reach the top of the hill from the bottom of the valley. Here we use Mountain Car to see if our approach can learn about more than just getting out of the valley quickly. We specified a GVF whose termination and policy focuses on reaching top of the left hill, and a second GVF about reaching the top of the other side. The full details of the GVFs and setup of this task can be found in the Appendix E.3.

Figure 5a shows how GPI and Sarsa compare against a baseline random policy. GPI provides much better data for GVF learning than the random policy and Sarsa, significantly reducing the RMSE of the learned GVFs. The goal visitation plots show GPI explores the domain and visits both GVFs goal far more often than random, and more effectively than Sarsa.

## 8   Conclusion

In this work, we take the first few steps towards building an effective multi-prediction learning system. We highlight the inherent non-stationarity in the problem and design algorithms based on successor features (SF) to better adapt to this non-stationarity. We show that (1) temporally consistent behavior emerges from optimizing the amount of learning across diverse GVF questions; (2) successor features are useful for tracking nonstationary rewards and cumulants, both in theory and empirically; (3) replay is well suited for learning the stationary components successor features while meta-learning works well for the non-stationary components; and (4) interest functions can improve the performance of the entire system, by focusing learning to a subset of states for each prediction.

Our work also highlights several critical open questions. (1) The utility of SFs is tied to the quality of the reward features; better understanding of how to learn these reward features is essential. (2) Continual Auxiliary Task Learning is an RL problem, and requires effective exploration approaches to find and maximize intrinsic rewards—the intrinsic rewards do not provide a solution to exploration. Never-ending exploration is needed. (3) The interaction between discovering predictive questions and learning them effectively remains largely unexplored. In this work, we focused on learning, for a given set of GVFs. Other work has focused on discovering useful GVFs [Veeriah et al., 2019, 2021, Nair et al., 2020, Zahavy et al., 2021]. The interaction between the two is likely to introduce additional complexity in learning behavior, including producing automatic curricula observed in previous work [Oudeyer et al., 2007, Chentanez et al., 2005].

This work demonstrates the utility of several new ideas in RL that are conceptually compelling, but not widely used in RL systems, namely SF and GVFs, GPI with SF for control, meta-descent step-size adaption, and interest functions. The trials and tribulations that lead to this work involved many failures using classic algorithms in RL, like replay; and, in the end, providing evidence for utility in these newer ideas. Our journey highlights the importance of building and analyzing complete RL systems, where the interacting parts—with different timescales of learning and complex interdependencies—necessitate incorporating these conceptually important ideas. Solving these integration problems represents the next big step for RL research.

## Acknowledgments and Disclosure of Funding

This work was supported by NSERC Discovery, IVADO, CIFAR through CCAI Chair funding and by the Alberta Machine Intelligence Institute (Amii).

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
