## A    Sample Efficiency of SF-NR

### A.1    Proof of Lemma 1

In this section we provide a proof of Lemma 1. The result is included for completeness, and similar results of a similar form can be found throughout the literature (for example, similar steps are used in the proof of Lemma 3 of Scherrer [2016]). The lemma is repeated for convenience below.

**Lemma 1** *Assume there exists a $\mathbf{w}^* \in \mathbb{R}^d$ such that $\mathbf{r}^\pi = \mathbf{X}\mathbf{w}^*$. Let $\hat{\mathbf{r}} \stackrel{def}{=} \mathbf{X}\mathbf{w}$ for some $\mathbf{w} \in \mathbb{R}^d$, and let $\mathbf{D} = Diag(\{d(s)\}_{s \in \mathcal{S}})$ for distribution $d$ fully supported on $\mathcal{S}$, with $\|\cdot\|_\mathbf{D}$ the weighted norm under $\mathbf{D}$. Then the value estimate $\hat{\mathbf{v}} \stackrel{def}{=} \Psi\mathbf{w}$ satisfies $\frac{1}{2}\|\mathbf{v}^\pi - \hat{\mathbf{v}}\|_\mathbf{D}^2 \leq \frac{\|\mathbf{r}^\pi - \hat{\mathbf{r}}\|_\mathbf{D}^2}{2(1-\gamma)^2}$.*

*Proof:* Given access to an estimate $\hat{\mathbf{r}}$ of $\mathbf{r}^\pi$ can then bound the MSVE as

$$
\begin{aligned}
\frac{1}{2}\|\mathbf{v}^\pi - \hat{\mathbf{v}}\|_\mathbf{D}^2 &\stackrel{(a)}{=} \frac{1}{2}\|(\mathbf{I} - \gamma\mathbf{P}_\pi)^{-1}(\mathbf{r}^\pi - \hat{\mathbf{r}})\|_\mathbf{D}^2 \\
&\stackrel{(b)}{\leq} \frac{1}{2}\|(\mathbf{I} - \gamma\mathbf{P}_\pi)^{-1}\|_\mathbf{D}^2 \|\mathbf{r}^\pi - \hat{\mathbf{r}}\|_\mathbf{D}^2 \\
&\stackrel{(c)}{\leq} \frac{1}{2}\left(\sum_{t=0}^\infty \gamma^t \|\mathbf{P}_\pi^t\|_\mathbf{D}\right)^2 \|\mathbf{r}^\pi - \hat{\mathbf{r}}\|_\mathbf{D}^2 \\
&\stackrel{(d)}{\leq} \frac{\|\mathbf{r}^\pi - \hat{\mathbf{r}}\|_\mathbf{D}^2}{2(1-\gamma)^2}
\end{aligned}
$$

where $(a)$ decomposed $\mathbf{v}^\pi = (\mathbf{I} - \gamma\mathbf{P}_\pi)^{-1}\mathbf{r}^\pi$, $(b)$ uses sub-multiplicativity of the matrix norm induced by $\|\cdot\|_D$, $(c)$ uses the von-neumann expansion $(\mathbf{I} - \gamma\mathbf{P}_\pi)^{-1} = \sum_{t=0}^\infty \gamma^t\mathbf{P}_\pi^t$ and triangle inequality, and $(d)$ uses that $\|\mathbf{P}_\pi^t\|_D = \lambda_{\max}\left((\mathbf{P}_\pi^t)^\top \mathbf{D}\mathbf{P}_\pi^t\right) \leq 1$ since both $\mathbf{D}$ and $\mathbf{P}_\pi^t$ have eigenvalues of at-most 1, followed by $\sum_{t=0}^\infty \gamma^t = \frac{1}{1-\gamma}$.

∎

### A.2    Proof of Proposition 1

**Proposition 1** *Define $\ell_t(w) \stackrel{def}{=} \frac{\rho_t}{2}(r_t - \langle x(S_t), w\rangle)^2$. Suppose we apply a basic recursive least-squares estimator to minimize regret on this loss sequence, producing a sequence of iterates $w_t$. Let $\overline{w}_T \stackrel{def}{=} \frac{1}{T}\sum_{t=1}^T w_t$ denote the average iterate. For $\hat{v}(s) = \langle \psi(s), \overline{w}_T\rangle$, we have that*

$$
\|\mathbf{v}^\pi - \hat{\mathbf{v}}\|_\mathbf{D}^2 \leq O\left(\frac{d\rho_{\max}R_{\max}^2 \log\left(1 + \rho_{\max}L^2 T\right)}{(1-\gamma)^2 T}\right). \tag{2}
$$

*Proof:*

$$
\begin{aligned}
\mathcal{L}(\overline{w}_T) &= \mathbb{E}_{d_\pi}\left[\frac{1}{2}(r_t - \langle x(S_t), \overline{w}_T\rangle)^2\right] \\
&= \mathbb{E}_{d_\mu}\left[\frac{\rho_t}{2}(r_t - \langle x(S_t), \overline{w}_T\rangle)^2\right] \\
&\leq \mathbb{E}_{d_\mu}\left[\frac{1}{T}\sum_{t=1}^T \ell_t(w_t)\right] \\
&\leq \frac{\|w^*\|^2 + d\rho_{\max}R_{\max}^2 \log\left(1 + \rho_{\max}L^2 T\right)}{2T},
\end{aligned}
$$

where in the last line we applied the regret guarantee of the RLS estimator with regularization parameter $\lambda = 1$ (See Orabona [2019, Theorem 7.26]) and used that $\max_s \|\phi(s)\|_2 \leq \sqrt{d}\max_s \|\phi(s)\|_\infty = \sqrt{d}L$. Following Lemma 1, by taking $\hat{v}(s) = \langle \psi(s), \overline{w}_T\rangle$, we have that

$$
\|\mathbf{v}^\pi - \hat{\mathbf{v}}\|_\mathbf{D}^2 \leq O\left(\frac{d\rho_{\max}R_{\max}^2 \log\left(1 + \rho_{\max}L^2 T\right)}{(1-\gamma)^2 T}\right). \tag{3}
$$

∎

# B Relationship between SF-NR and TD solutions

Let $\mathbf{w}^\pi$ be the fixed-point solution for the projected Bellman operator with respect to the $\lambda$-return that is be estimated by LSTD($\lambda$) for policy $\pi$[White, 2017]. It is well known that TD($\lambda$) converges to this solution under the right conditions. The components of the solution $\mathbf{w}^\pi = \mathbf{A}_\pi^{-1}\mathbf{b}_\pi$ are as follows

$$\mathbf{A}_\pi = \mathbf{X}^\top \mathbf{D}(\mathbf{I} - \lambda\mathbf{P}_\gamma\mathbf{\Pi}_\pi)^{-1}(\mathbf{I} - \mathbf{P}_\gamma\mathbf{\Pi}_\pi)\mathbf{X}$$

$$\mathbf{b}_\pi = \mathbf{X}^\top \mathbf{D}(\mathbf{I} - \lambda\mathbf{P}_\gamma\mathbf{\Pi}_\pi)^{-1}\mathbf{r}$$

where $\mathbf{X} \in \mathbb{R}^{|\mathcal{S}||\mathcal{A}|\times d}$ is the feature matrix with $\mathbf{x}(s,a)^\top$ along its rows, $\mathbf{r} \in \mathbb{R}^{|\mathcal{S}||\mathcal{A}|}$ is the expected immediate reward $\left(\mathbf{r}((s,a)) = \sum_{s'\in\mathcal{S}} P(s,a,s')R(s,a,s')\right)$, $\mathbf{P}_\gamma \in \mathbb{R}^{|\mathcal{S}||\mathcal{A}|\times|\mathcal{S}|}$ is a sub-stochastic matrix that represents the transition process $\left(\mathbf{P}_\gamma((s,a),s') = P(s,a,s')\gamma(s,a,s')\right)$, $\mathbf{\Pi}_\pi \in \mathbb{R}^{|\mathcal{S}|\times|\mathcal{S}||\mathcal{A}|}$ is a stochastic matrix that represents $\pi$ $\left(\mathbf{\Pi}_\pi(s,(s,a)) = \pi(s,a)\right)$, and $\mathbf{D} \in \mathbb{R}^{|\mathcal{S}|\times|\mathcal{S}||\mathcal{A}|}$ is a diagonal matrix with the stationary distribution induced by $\pi$ on its diagonal that controls the approximation error.

Let us consider $\lambda = 1$ case for simplicity. Under this case

$$\mathbf{A}_\pi = \mathbf{X}^\top\mathbf{D}\mathbf{X}$$

$$\mathbf{b}_\pi = \mathbf{X}^\top\mathbf{D}(\mathbf{I} - \mathbf{P}_\gamma\mathbf{\Pi}_\pi)^{-1}\mathbf{r}$$

The predicted values correspond to $\hat{\mathbf{Q}} = \mathbf{X}\mathbf{w}^\pi$. This is the projection of the true values $\mathbf{Q}^* = (\mathbf{I} - \mathbf{P}_\gamma\mathbf{\Pi}_\pi)^{-1}\mathbf{r}$ onto the space spanned by $\mathbf{X}$ where the projection operator is $\mathbf{\Pi} = \mathbf{X}(\mathbf{X}^\top\mathbf{D}\mathbf{X})^{-1}\mathbf{X}^\top\mathbf{D}$ – the TD(1) solution. Now, depending on if the form of the reward $\mathbf{r}$ we have the three following cases.

*Case 1:* $\mathbf{r} = \mathbf{X}\mathbf{w}$ The LSTD estimate can be written as

$$\theta_{\text{LSTD}} = (\mathbf{X}^\top\mathbf{D}\mathbf{X})^\top\mathbf{X}^\top\mathbf{D}(\mathbf{I} - \mathbf{P}_\gamma\mathbf{\Pi}_\pi)^{-1}\mathbf{X}\mathbf{w}$$

where the component $\mathbf{\Psi} = (\mathbf{I} - \mathbf{P}_\gamma\mathbf{\Pi}_\pi)^{-1}\mathbf{X}$ corresponds to the successor features. Therefore, if the space $\mathbf{X}$ is used for learning both the successor features and the reward, the solution corresponding to SF-NR would be equivalent to the solution obtained by TD.

*Case 2:* $\mathbf{r} = \mathbf{\Phi}\mathbf{w}$ The LSTD estimate can be written as

$$\theta_{\text{LSTD}} = (\mathbf{X}^\top\mathbf{D}\mathbf{X})^\top\mathbf{X}^\top\mathbf{D}(\mathbf{I} - \mathbf{P}_\gamma\mathbf{\Pi}_\pi)^{-1}\mathbf{\Phi}\mathbf{w}$$

where the component $\mathbf{\Psi} = (\mathbf{I} - \mathbf{P}_\gamma\mathbf{\Pi}_\pi)^{-1}\mathbf{\Phi}$ corresponds to the successor features. Therefore, if the space $\mathbf{X}$ is used for learning the successor features which correspond to weighted sums of $\mathbf{\Phi}$, and $\mathbf{\Phi}$ is used for learning the reward, the solution corresponding to SF-NR would be equivalent to the solution obtained by TD.

*Case 3:* $\mathbf{r} = \mathbf{X}\mathbf{w} + \eta_{\mathbf{r}}$ *or* $\mathbf{r} = \mathbf{\Phi}\mathbf{w} + \eta_{\mathbf{r}}$ where $\eta_{\mathbf{r}}$ is the model misspecification error for predicting the immediate reward, the LSTD estimate would correspond to

$$\theta_{\text{LSTD}} = (\mathbf{X}^\top\mathbf{D}\mathbf{X})^\top\mathbf{X}^\top\mathbf{D}(\mathbf{I} - \mathbf{P}_\gamma\mathbf{\Pi}_\pi)^{-1}\mathbf{r}$$

whereas the SF-NR estimate would capture the same component as in case (1), or in case (2). Therefore, if there is a misspecification error for learning $\mathbf{r}$, the two solutions would differ.

Hence, decomposition of SF-NR does not reduce representability of the TD(1) solution if the reward is linearizable in some features. More generally, we could introduce $\lambda < 1$ to provide.a bias-variance trade-off for learning the SR as well.

# C Prior Corrections and the Projected Bellman Error

Let us first consider the SR objective under a fixed behavior, $\mu$, with stationary distribution $d_\mu$ over states and actions. When using TD for action-values, with covariance $\mathbf{C} = \mathbb{E}[\mathbf{x}(S,A)\mathbf{x}(S,A)^\top] = \sum_{s,a} d_\mu(s,a)\mathbf{x}(S,A)\mathbf{x}(S,A)^\top$, the underlying objective is the mean-squared projected Bellman error (MSPBE):

$$\text{MSPBE}(w) = \|\sum_{s,a} d_\mu(s,a)\mathbb{E}_\pi[\delta(w)\mathbf{x}(s,a)|S=s,A=a]\|^2_{\mathbf{C}^{-1/2}}$$

$$= \mathbb{E}_\pi[\delta(w)\mathbf{x}(S,A)]^\top\mathbf{C}^{-1}\mathbb{E}_\pi[\delta(w)\mathbf{x}(S,A)]$$

The TD fixed point corresponds to $w$ such that $\mathbb{E}_\pi[\delta(w)\mathbf{x}(S, A)] = 0$, which is defined based on state-action weighting $d_\mu$. Different weightings result in different solutions.

The weighting is implicit in the TD update, when updating from state and actions visited under the behavior policy. The predictions are updated more frequently in the more frequently visited state-action pairs, giving them higher weighting in the objective. However, we can change the weighting using important sampling. For example, if we pre-multiply the TD update with $d(s, a)/d_\mu(s, a)$ for some weighting $d$, then this changes the state-action weighting in the objective to $d(s, a)$ instead of $d_\mu(s, a)$.

The issue, though, is not that the objective is weighted by $d_\mu$, but rather that $d_\mu$ is changing as $\mu$ is changing. Correspondingly, the optimal SR solution could be changing since the objective is changing. The impact of this changing state distribution depends on the function approximation capacity. The weighting indicates how to trade-off function approximation error across states; when approximation error is low or zero, the weighting has no impact on the TD fixed point. For example, in a tabular setting, the agent can achieve $\mathbb{E}_\pi[\delta(w)\mathbf{x}(s, a)|S = s, A = a] = 0$ for every $(s, a)$. Regardless of the weighting—as long as it is non-zero—the TD fixed point is the same.

Generally, however, there will be some approximation error and so some level of non-stationarity. This pre-multiplication provides us with a mechanism to keep the objective stationary. If we could track the changing $d_{\mu_t}$ with time, and identify a desired weighting $d$, then we could pre-multiply each update with $d(s_t, a_t)/d_{\mu_t}(s_t, a_t)$ to ensure we correct the state-action distribution to be $d$. There have been some promising strategies developed to estimate a stationary $d_\mu$ [Hallak and Mannor, 2017, Liu et al., 2018b, 2020], though here they would have to be adapted to constantly track $d_{\mu_t}$.

Another option is to use prior corrections to reweight the entire trajectory up to a state. Prior corrections were introduced to ensure convergence of off-policy TD [Precup, 2000]. For a fixed behavior, the algorithm pre-multiplies with a product of important sampling ratios, with $\rho(a|s) \stackrel{\text{def}}{=} \frac{\pi(a|s)}{\mu(a|s)}$

$$w = w + \alpha \left[ \Pi_{i=0}^t \rho(a_i|s_i) \right] \delta\mathbf{x}(s_t, a_t)$$

This shifts the weight from state-actions visited under $\mu$ to state-actions visited under $\pi$, because

$$\mathbb{E}_\mu \left[ \Pi_{i=0}^t \rho(A_i|S_i)\delta\mathbf{x}(S_t, A_t)|S_t = s, A_t = a \right]$$
$$= \mathbb{E}_\mu \left[ \Pi_{i=0}^t \rho(A_i|S_i)|S_t = s, A_t = a \right] \mathbb{E}[\delta\mathbf{x}(S_t, A_t)|S_t = s, A_t = a]$$

and when considering expectation across time steps $t$ when $s, a$ are observed

$$\mathbb{E}_\mu \left[ \Pi_{i=0}^t \rho(A_i|S_i)|S_t = s, A_t = a \right] = \frac{d_\pi(s, a)}{d_\mu(s, a)}$$

These prior corrections also corrects the state-action distribution even with $d_\mu$ changing on each step, because the numerator reflects the probability of reach $s, a$ under policy $\pi$ and the denominator reflects the probability of reach $s, a$ using the sequence of behavior distributions. For $\rho_t(a|s) \stackrel{\text{def}}{=} \frac{\pi(a|s)}{\mu_t(a|s)}$

$$\Pi_{i=0}^t \rho_t(A_i|S_i) = \frac{\pi(A_0|S_0)\pi(A_1|S_1)\dots\pi(A_t|S_t)}{\mu_0(A_0|S_0)\mu_1(A_1|S_1)\dots\mu_t(A_t|S_t)}$$
$$= \frac{\pi(A_0|S_0)P(S_1|S_0, A_0)\dots P(S_t|S_{t-1}, A_{t-1})\pi(A_t|S_t)}{\mu_0(A_0|S_0)P(S_1|S_0, A_0)\dots P(S_t|S_{t-1}, A_{t-1})\mu_t(A_t|S_t)}$$

# D   Algorithms

The algorithm for Tree-Backup($\lambda$) is from Precup [2000].

---
**Algorithm 3** TB($\lambda$) Update
---
$\mathbf{z}_t = \gamma_t \pi(A_t|S_t)\lambda\mathbf{z}_{t-1} + \mathbf{x}(S_t, A_t)$
$\delta_t = c_t + \gamma_{t+1}\sum_{a'}\pi(a'|S_{t+1})\hat{q}(S_{t+1}, a') - \hat{q}(S_t, A_t)$
$\mathbf{w}_{t+1} = \mathbf{w}_t + \eta_t\delta_t\mathbf{z}_t$

---

Algorithm 4 is the online TB with interest update. The derivation for the online update rule from the forward view is in the next section.

---

**Algorithm 4** TB($\lambda$) with Interest Update

$\mathbf{z}_t = \gamma_t \pi(A_t|S_t)\lambda \mathbf{z}_{t-1} + I_t \mathbf{x}(S_t, A_t)$
$\delta_t = c_t + \gamma_{t+1} \sum_{a'} \pi(a'|S_{t+1})\hat{q}(S_{t+1}, a') - \hat{q}(S_t, A_t)$
$\mathbf{w}_{t+1} = \mathbf{w}_t + \eta_t \delta_t \mathbf{z}_t$

---

The ETB($\lambda$) algorithm is from Sutton et al. [2016] ETD($\lambda$) but modified to use TB($\lambda$) update instead of TD($\lambda$) update. This modification relies on the correspondence between TB and TD, where TB is a version of TD with the variable trace parameter, $\lambda_t = b(a_t|s_t)\lambda$. This correspondence is demonstrated in [Mahmood et al., 2017, Ghiassian et al., 2018]. Here, we use the same replacement of $\lambda_t$ in ETD($\lambda$) to get ETB($\lambda$).

---

**Algorithm 5** Emphatic TB($\lambda$) Update

$F_t = \rho_{t-1}\gamma_t F_{t-1} + I_t$
$M_t = \rho_t\Big[\lambda b(A_t|S_t)I_t + \big(1 - \lambda b(A_t|S_t)\big)F_t\Big]$
$\mathbf{z}_t = \gamma_t \pi(A_t|S_t)\lambda \mathbf{z}_{t-1} + M_t \mathbf{x}(S_t, A_t)$
$\delta_t = c_t + \gamma_{t+1} \sum_{a'} \pi(a'|S_{t+1})\hat{q}(S_{t+1}, a') - \hat{q}(S_t, A_t)$
$\mathbf{w}_{t+1} = \mathbf{w}_t + \eta_t \delta_t \mathbf{z}_t$

---

### D.1 Online Interest TB Derivation

The forward view update that uses interest at each time-step is of the form

$$\mathbf{w}_{t+1} = \mathbf{w}_t + \alpha I_t(G_t - \hat{q}(S_t, A_t, \mathbf{w}_t))\nabla\hat{q}(S_t, A_t, \mathbf{w}_t).$$

According to Sutton and Barto [2018] (page 313), ignoring the changes in the approximate value function, the TB return can be written as,

$$G_t \approx \hat{q}(S_t, A_t, \mathbf{w}_t) + \sum_{k=t}^{\infty} \delta_k \prod_{i=t+1}^{k} \gamma_i \lambda_i \pi(A_i|S_i). \tag{4}$$

We substitute Equation 4 for $G_t$ in the forward view update, we get,

$$\mathbf{w}_{t+1} \approx \mathbf{w}_t + \alpha I_t \sum_{k=t}^{\infty} \delta_k \prod_{i=t+1}^{k} \gamma_i \lambda_i \pi(A_i|S_i)\nabla\hat{q}(S_t, A_t, \mathbf{w}_t).$$

The sum of forward view update over time is

$$\sum_{t=1}^{\infty}(\mathbf{w}_{t+1} - \mathbf{w}_t) \approx \sum_{t=1}^{\infty}\sum_{k=1}^{\infty} \alpha I_t \delta_k \nabla\hat{q}(S_t, A_t, \mathbf{w}_t) \prod_{i=t+1}^{k} \gamma_i \lambda_i \pi(A_i|S_i)$$

$$= \sum_{k=1}^{\infty}\sum_{t=1}^{k} \alpha I_t \nabla\hat{q}(S_t, A_t, \mathbf{w}_t)\delta_k \prod_{i=t+1}^{k} \gamma_i \lambda_i \pi(A_i|S_i)$$

$$= \sum_{k=1}^{\infty} \alpha\delta_k \sum_{t=1}^{k} I_t \nabla\hat{q}(S_t, A_t, \mathbf{w}_t) \prod_{i=t+1}^{k} \gamma_i \lambda_i \pi(A_i|S_i).$$

This can be a backward-view TD update if the entire expression from the second sum can be estimated incrementally as an eligibility trace. Therefore

$$
\begin{aligned}
\mathbf{z}_k &= \sum_{t=1}^{k} I_t \nabla \hat{q}(S_t, A_t, \mathbf{w}_t) \prod_{i=t+1}^{k} \gamma_i \lambda_i \pi(A_i | S_i) \\
&= \sum_{t=1}^{k-1} I_t \nabla \hat{q}(S_t, A_t, \mathbf{w}_t) \prod_{i=t+1}^{k} \gamma_i \lambda_i \pi(A_i | S_i) + I_k \nabla \hat{q}(S_k, A_k, \mathbf{w}_k) \\
&= \gamma_k \lambda_k \pi(A_k | S_k) \sum_{t=1}^{k-1} I_t \nabla \hat{q}(S_t, A_t, \mathbf{w}_t) \prod_{i=t+1}^{k-1} \gamma_i \lambda_i \pi(A_i | S_i) + I_k \nabla \hat{q}(S_k, A_k, \mathbf{w}_k) \\
&= \gamma_k \lambda_k \pi(A_k | S_k) \mathbf{z}_{k-1} + I_k \nabla \hat{q}(S_k, A_k, \mathbf{w}_k).
\end{aligned}
$$

Changing the index from $k$ to $t$, the accumulating trace update can be written as,

$$
\mathbf{z}_t = \gamma_t \lambda_t \pi(A_t | S_t) \mathbf{z}_{t-1} + I_t \nabla \hat{q}(S_t, A_t, \mathbf{w}_t),
$$

leading to the incremental update for estimating $\mathbf{w}_{t+1}$.

### D.2   Auto Optimizer

We use a variant of the Autostep optimizer throughout our experiments. Adam and RMSProp are global update scaling methods and do note adapt step-sizes on a per feature basis [Kingma and Ba, 2015], unlike meta descent methods like IDBD, Autostep [Mahmood et al., 2012], and AdaGain [Jacobsen et al., 2019]—this is critical for achieving introspective learners. Meta-descent methods like Autostep have been shown to be very effective with linear function approximation [Jacobsen et al., 2019]. Jacobsen's AdaGain algorithm is rather complex, requiring finite differencing, whereas Auto is a simple method that works nearly as well in practice. In our own preliminary experiments, we found Adam to much less effective at tracing non-stationary learning targets, even when we adapted all three hyperparameters of the method. Finally, Auto can be seen as optimizing a meta objective for the step-size and thus is a specialization of Meta-RL to online step-size adaption in RL.

There have been attempts to apply the Autostep algorithm to TD and Sarsa [Dabney and Barto, 2012]. Auto represents another attempt to use Autostep in the reinforcement learning setting. Modifications to the Autostep algorithm are from personal communications with an author of the original work [Mahmood et al., 2012] on how to make it more effective in practice in the reinforcement learning setting.

---

**Algorithm 6** Auto Update

$\mathbf{n} = \mathbf{n} + \frac{1}{\tau} \boldsymbol{\alpha} |\phi| \cdot (|\mathbf{h} \cdot \delta\phi| - \mathbf{n})$
**for all** i such that $\phi_i \neq 0$ **do**

$\quad \triangle\beta_i = \text{clip}\left( -M_\triangle, \left| \dfrac{\mathbf{h}_i \delta\phi_i}{\mathbf{n}_i} \right| \right)$

$\quad \boldsymbol{\alpha}_i = \text{clip}(\kappa, \boldsymbol{\alpha}_i e^{\mu \triangle\beta_i}, \dfrac{1}{|\phi_i|})$

**end**
**if** $\boldsymbol{\alpha}^T \mathbf{z} > 1$

$\quad \forall i$ such that $\mathbf{z}_i \neq 0$: $\boldsymbol{\alpha}_i = \min(\boldsymbol{\alpha}_i, \dfrac{1}{|\mathbf{z}|_1})$

**end**
$\boldsymbol{\theta} = \boldsymbol{\theta} + \boldsymbol{\alpha} \cdot \delta\phi$
$\mathbf{h} = \mathbf{h}(1 - \boldsymbol{\alpha} \cdot |\phi|) + \boldsymbol{\alpha}\dot{\delta\phi}$

---

where:

- $\mu$ is the meta-step size parameter.
- $\boldsymbol{\alpha}$ is the step sizes.
- $\delta$ is the scalar error.
- $\phi$ is the feature vector.
- $\mathbf{z}$ is the step-size truncation vector.
- $\boldsymbol{\theta}$ is the weight vector.
- $\mathbf{h}$ is the decaying trace.
- $\mathbf{n}$ maintains the estimate of $|\mathbf{h} \cdot \delta\phi|$.
- $\tau$ is the step size normalization parameter.
- $M_\triangle$ is the maximum update parameter of $\boldsymbol{\alpha}_i$.
- $\kappa$ is the minimum step size.

In all experiments, $M_\triangle = 1$, $\tau = 10^6$, $\kappa = 10^{-6}$. In the reinforcement setting, $\phi$ is the eligibility trace, $\delta$ is the td error and $\mathbf{z}$ is the overshoot vector. $\mathbf{z}$ is calculated as $|\phi| \cdot \max(|\phi|, |\mathbf{x} - \gamma\mathbf{x}'|)$, where $\mathbf{x}$ is the state representation at timestep $t$ and $\mathbf{x}'$ is the state representation at timestep $t + 1$.

# E  Experiment Details

This section provides additional details about the experiments in the main body, and the additional experiments in this appendix. All the experiments in this work used a combined compute usage of approximately five CPU months.

## E.1  TMaze Details

Tabular TMaze is a deterministic gridworld with four actions {up, down, left, right}. There are four GVFs being learned and each correspond to a goal as depicted in Figure 1. For GVF $i$ and the corresponding goal state $G_i$, $pi_i$, $\gamma_i$ and $c_i$ are defined as:

- $\pi_i(a|s)$: deterministic policy that directs the agent towards $G_i$
- $\gamma_i(G_i) = 0$, $\gamma_i(s) = 0.9 \; \forall s \neq G_i \in \mathcal{S}$
- $c_i^t(s, a, s') = \begin{cases} 0 & s' \neq G_i \\ C_i^t & s' = G_i \end{cases}$

where $C_i^t$ can be one of the following four different and possibly non-statioanry cumulant schedules:

- Constant: $C_i^t = C_i$
- Distractor: $C_i^t = N(\mu_i, \sigma_i)$
- Drifter: $C_i^t = C_i^{t-1} + N(\mu_i, \sigma_i), C_i^0 = 1$

As discussed in Section 3, the cumulants of the GVFs can be stationary or non-stationary signals. The cumulant of each GVF has a non-zero value at their respective goal. In the Tabular TMaze, the top left goal is a *distractor* cumulant which is an unlearnable noisy signal. The *distractor* has $\mu = 1$ and $\sigma^2 = 25$. The cumulants corresponding to the lower left goal and upper right goal are *constant* goals uniformly selected at the start of each run between $[-10, 10]$. The cumulant corresponding to the lower left goal is a *drifter* signal of $\sigma^2 = 0.01$ and represents a learnable non-stationary signal.

The *Continuous TMaze* follows the same design as the Tabular TMaze except it is embedded in a continuous 2D plane between 0 and 1 on both axes. Each hallway is a line with no width, allowing the agent to go along the hallway, but not perpendicular to it. The main vertical hallway spans between [0, 0.8] on the y-axis and is located at x of 0.5. The main horizontal hallway spans [0, 1] on the x axis and is located at y of 0.8. Finally, the two vertical side hallways span between [0.6, 1.0]. Junctions and goal locations occupy a $2\epsilon$ x $2\epsilon$ space at the end of each hallway. For example, the middle junction spans x of $0.5 \pm \epsilon$ and y of $0.8 \pm \epsilon$. The agent can take one of four actions four actions {up, down, left, right}. The agent moves in the corresponding direction of the action with a step size of 0.08 and noise altering movement by Uniform$(-0.01, 0.01)$. Figure 6 summarizes the environment set-up.

**Reward features**

As discussed in the main paper, SF-NR requires a reward feature, $\phi(s, a, s')$. In the Tabular TMaze, the reward features for both the GVFs and the behavior learner are the tabular representation of $\phi(s, a, s')$. For the Continuous TMaze, the reward features using SF-NR for the GVF learners is an indicator function for if in the tuple $(s, a, s')$ is in the GVF's goal state. Since the Continuous TMaze has four goals, $\phi$ for the GVF learners is a four dimensional vector. This is a reasonable feature vector as the reward feature vector should be related to rewarding transitions. For GPI, it is unclear what is a rewarding transition apriori. Therefore, the reward feature is the action-feature vector of state-aggregation applied to the Continuous TMaze. This is a general yet compact feature representation. Each line segment for the Continuous TMaze is broken up into thirds and state aggregation is applied to each part.

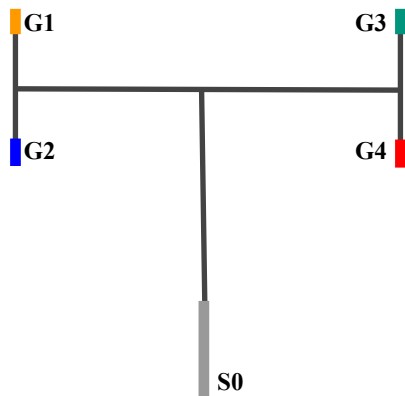

Figure 6: Continuous TMaze with the 4 GVFs. $S_0$, the grey shaded region, is the uniformly weighted starting state distribution after a goal visit.

**Algorithm parameters**

For the fixed behavior experiment in Tabular TMaze, the GVFs using TB($\lambda$) learners and SF-NR learners had their meta-step size swept from $[5^{-4}, ..., 5^0]$ and initial step size tested for $[0.1, 1.0]$. For both TB($\lambda$) and SF-NR, the optimal meta-step size was $5^{-1}$ and initial step size of $1.0$.

For the learned behavior experiment, the behavior learner and GVF learner share the same meta-step size parameter and initial alpha. The meta-step size was swept from $[5^{-4}, ..., 5^0]$ and initial step sizes were $[0.1, 1.0]$. All four agents (GPI behavior learner with TB($\lambda$) or SF-NR GVF learners, and SARSA behavior learner with TB($\lambda$) or SF-NR GVF learners) had an optimal meta-step size of $5^{-2}$. For agents using SF-NR GVF learners, the optimal initial step size was $1.0$. For agents using TB($\lambda$) GVF learners, the initial step size was $0.1$. The behavior learner is optimistically initialized to ensure that the agent will visit each of the four goals at least once. To the best of our knowledge, no one has tried optimistic initialization with successor features before. To perform optimistic initialization for GPI, we initialized all successor features, $\psi$, estimate to be $\mathbf{1}$. We initialized the immediate reward estimates, $\mathbf{w}$, to the desired optimistic initialization threshold normalized by the number of reward features. We believe this to be an approximate version of optimistic initialization to allow comparisons to the SARSA agent. All behaviors used a fixed $\epsilon$ of $0.1$ for the runs. The agent's performance was evaluated on the TE for the last 10% of the runs.

For the fixed behavior experiment in Continuous TMaze, the meta-step size parameter was swept from $[5^{-3}, ..., 5^0]$ and the initial step size was swept over $[0.01, 0.1, 0.2]$. The initial step size was then divided by the number of tilings of the tile coder to ensure proper scaling. For the fixed behavior experiment, the optimal meta-step size for SF-NR and TB($\lambda$) GVF learners was $5^{-2}$ with an initial step size of $0.2$.

For the learned behavior experiment in Continuous TMaze, the behavior learners and GVF learners shared the same meta-step size parameter and initial step size. The meta-step size parameter was swept from $[5^{-4}, ..., 5^0]$ and the initial step size was swept over $[0.01, 0.1, 0.2]$. For GPI, the optimal initial step size for both types of GVF learners was $0.2$. For SF-NR learners, the optimal meta-step size was $5^{-3}$ while being $5^{-2}$ for the TB($\lambda$) GVF learners. For the Sarsa behavior learner, the optimal meta-step size was $5^{-2}$ and the initial step size when the SF-NR GVF learners were used was $0.2$ while being $0.1$ for the TB($\lambda$) learners. The behavior learner was optimistically initialized and used an $\epsilon$ of $0.1$. The agent's performance was evaluated on the TE for the last 10% of the runs.

Since the intrinsic reward (weight change) is $\geq 0$, the intrinsic reward is augmented with a modest $-0.01$ reward per step to encourage the agent to seek out new experiences.

### E.2 Open 2D World Details

The Open 2D World is an open continuous grid world with boundaries defined by a square of dimensions $10 \times 10$, with goals in each of the four corners. The goals follow the same schedules as defined for the TMaze experiments with *constants* sampled from $[-10, 10]$, *drifter* parameters of $\sigma^2 = 0.005$ and the initial value of 1, and *distractor* parameters of $N(\mu = 1, \sigma^2 = 1)$. On each step, the agent can select between the four compass actions. This moves the agent 0.5 units in the chosen direction with uniform noise $[-0.1, 0.1]$. A uniform $[-0.001, 0.001]$ orthogonal drift is also applied. The goals are squares in the corners with a size of $1 \times 1$. The start state distributions is the center of the environment $(x, y) \in [0.45, 0.55]^2$. The summary of the environment is shown in Figure 7. Similar to the TMaze variants, the GVF policies are defined as the shortest path to their respective goal. When there are multiple actions at a state that are part of a shortest path, these actions are equally weighted. The discount for the GVFs is $\gamma = 0.95$ for all states other than the goal states.

**Reward features**

The GVF reward features for SF-NR are defined similarly to the reward features in TMaze as described in Appendix E.1. Since there are four goals, $\phi \in \mathbf{R}^4$, where the value at $\phi_i$ is the indicator function for if $s' \in G_i$. This is a reasonable feature as it is focused on rewarding events. For the reward features of GPI, state aggregation is applied with tiles of size (2,2) and is augmented to be a state-action feature.

**Algorithm parameters**

The meta-step size for the behavior learners and the GVF learners were swept independently. The behavior learner's meta-step size was swept over $[5^{-5}, ..., 5^0]$ while the GVF learner's meta-step size was swept over $[5^{-4}, ...5^0]$. An initial step size of 0.1 scaled down by the number of tilings was used for all learners. The optimal meta-step size for the GVF learners that used ETB($\lambda$) was $5^{-3}$ and the meta-step size for the corresponding GPI learner was $5^{-1}$. Variance was an issue for learning successor features through ETB($\lambda$) so the emphasis was clipped at 1. For the agents using TB with Interest, the optimal meta-step size for the behavior learner was $5^{-1}$ and for the GVF learners was $5^{-2}$. For the method using no prior corrections, the behavior learner used a meta-step size of $5^{-4}$ and the GVF learner used a meta-step size of $5^{-2}$.

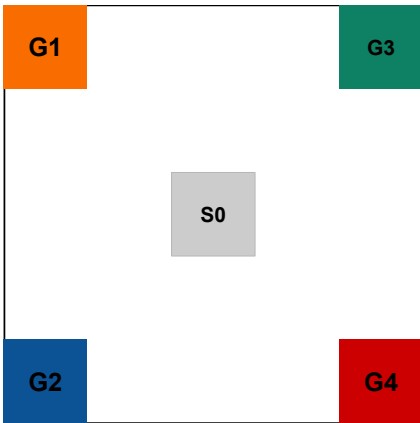

Figure 7: Open 2D World with the 4 GVFs situated at each corner. $S_0$, the grey shaded region, is the uniformly weighted starting state distribution after a goal visit.

Since the intrinsic reward (weight change) is $\geq 0$, the intrinsic reward is augmented with a modest $-0.05$ reward per step to encourage the agent to seek out new experiences.

### E.3 Mountain Car Details

We use the standard mountain car environment [Sutton and Barto, 2018] defined through a system of equations

$$A_t \in [\text{Reverse} = -1, \text{Neutral} = 0, \text{Throttle} = 1]$$
$$\dot{x}_{t+1} = \dot{x}_t + 0.001 A_t - 0.0025 \cos(3x_t)$$
$$x_{t+1} = x_t + \dot{x}_{t+1}.$$

We define two GVFs as auxiliary tasks. The first GVF receives a non-zero cumulant of value 1 when the agent reaches the left wall. It has a discount $\gamma = 0.99$ that terminates when the left wall is touched, and a policy that is learned offline to maximize the cumulant. The second GVF is similar, but receives a non-zero cumulant of 1 when reaching the top of the hill (the typical goal state). Each policy is learned offline for 500k steps on a transformed problem of the cumulant being -1 per step with ESARSA($\lambda$) and an $\epsilon = 0.1$. This allows a denser reward signal for learning a high quality policy. Note that the final policy for maximizing this cumulant signal and the sparse reward signal are the same. The state representation used for these policies is an independent tile coder of 16 tilings

with 2 tiles per dimension. The fixed policy after offline learning for each of the GVFs is greedy with respect to the learned offline action values.

**Reward features**

The reward features for the SF-NR learners are defined similarly to the reward features in the Continuous TMaze. $\phi(s, a, s')_i$ is 1 if and only if $s'$ is in the termination zone of GVF$_i$. The reward feature for GPI is a tile coder of 8 tilings with 2 tiles per dimension.

**Algorithm parameters**

The SF-NR and behavior learner are optimized with stochastic gradient descent in an online learning setting. The behavior step size and GVF step size were swept independently at $[3^{-2}, ...3^0]$. The values were then divided by the number of tilings to ensure proper scaling of step size. The behavior was epsilon-greedy with $\epsilon$ swept over the range $[0.1, 0.3, 0.5]$. For both GPI and Sarsa, $\epsilon = 0.1$ performed the best. For agents using GPI as the behavior learner, the optimal behavior step size was $3^0$ with the optimal GVF learner step size of $3^{-2}$. For the agents using a Sarsa behavior learner, the optimal parameters were a step size of $3^0$ for the behavior learner and a step size of $3^{-1}$ for the GVF learner. For the baseline agent where the behavior was random actions, the optimal GVF step size was $3^{-1}$.

Since the intrinsic reward (weight change) is $\geq 0$, the intrinsic reward is augmented with a modest $-0.01$ reward per step to encourage the agent to seek out new experiences.

# F    Additional Experiments

## F.1    Goal Visitation in Continuous TMaze

Figure 8 shows the goal visitation plots for in the Continuous TMaze for GPI and Sarsa. Using either GPI or SF-NR results in significantly faster identification of the *drifter* signal and together results in the fastest identification. Prioritizing visiting the drifter is the preferred behavior as it is the only learnable signal. It is important that the agent does not get confused by the *distractor* as it is not a learnable signal.

## F.2    Goal Visitation in Tabular TMaze

Figure 9 shows the goal visitation plots for the Tabular TMaze.

## F.3    The Effect of Generalization in the Reward Features

GPI is sensitive to the reward feature representation that is used. Figure 10 shows what happens when the reward features of 8 tilings with 2x2 tiles are used in the Continuous TMaze for GPI. The agents are swept over the same interval of hyperparameters as described in Appendix E.1 for the Continuous TMaze. When this reward feature was used, the optimal meta-step size was $5^{-3}$. The initial step size was $0.2$ which was then scaled down by the number of tilings. GPI, with this reward features, was unable to learn an effective policy to reduce the TE in the Continuous TMaze. The tile coded reward feature representation has approximately three times the number of features than the handcrafted feature representation, yet it results in much worse performance. This highlights the need for the reward features to be learnable by an algorithm rather than being predefined.

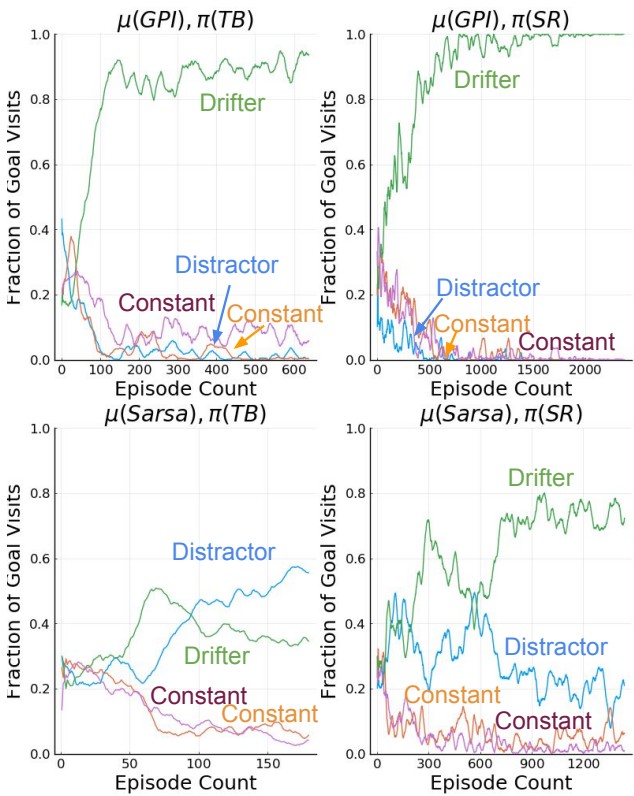

Figure 8: Goal visitation for GPI and Sarsa for the GVF learners using SF-NR and TB in Continuous TMaze. Episodes count are shown for the first N episodes up to the minimum number of episodes per run for each algorithm.

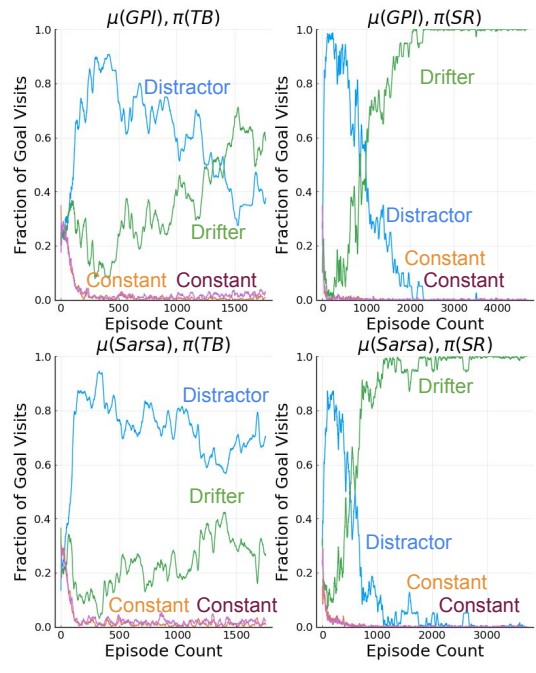

Figure 9: Goal visitation for Sarsa with the GVF learners using SF-NR and TB in Tabular TMaze.

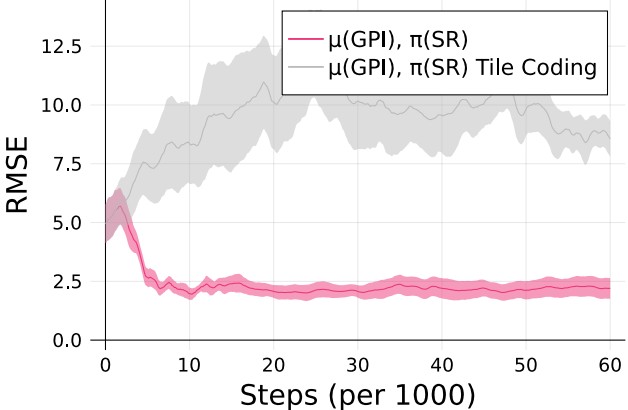

Figure 10: GPI with the same input features, but different reward features in the Continuous TMaze environment. $\mu(GPI), \pi(SR)$ uses the reward features detailed in Appendix E.1. $\mu(GPI), \pi(SR)$ Tile Coding uses the reward features of a tilecoder with 8 tilings of 2 tiles.