# OpenReview forum: "Continual Auxiliary Task Learning"
_NeurIPS.cc/2021/Conference — NeurIPS 2021 Poster_

### Official Review · Reviewer_WNTT · 2021-07-14

**Rating:** 7
**Confidence:** 3

**Summary:**

This paper studies the continual RL setting in which an agent behaves according to a policy designed to improve predictions on a number of auxiliary tasks. It proposes an algorithm based on successor features that facilitates tracking under non-stationary rewards, demonstrating both empirical and theoretical benefits of this approach.

**Main Review:**

This paper studies an important problem, namely that of learning about the environment without external rewards, in a very challenging setting, namely continual RL with inherent non-stationary for both the prediction learners and the behavior learner. The paper is well written, it introduces a new problem formulation, and proposes a new method, which is thoroughly evaluated empirically and analyzed from a theoretical perspective. I think this paper could be of value for the community and open up multiple directions for future work.

I particularly appreciated that the authors report results over 30 different runs of the models, which increases confidence in the drawn conclusions. I also liked the fact that the experiments were designed in simple environments, where it is easier to understand the strengths and limitations of these methods.

One limitation of this work is that the approach seems quite similar to prior work which uses successor features for transfer to new tasks (e.g. Barreto et al. 2017, Barreto et al. 2018 Borsa et al. 2018, Ma et al. 2020). Hence, the algorithmic contribution is not particularly novel, but I don't think this should prevent acceptance given that the empirical evaluation is quite thorough, the problem setting is novel and important, and the results look promising. However, I do think the authors should discuss in greater detail how the approach differs from prior work.

Another potential limitation of this work is the reliance upon the choice of policies / tasks to learn. I think it would be valuable to analyze or discuss how the algorithm depends on this choice and how one should choose the set of auxiliary tasks or policies. What if one doesn't have access to such tasks or policies -- how could these be discovered so that they are (most) useful for learning about the environment or downstream tasks without prior knowledge? How could one combine this with skill discovery or other related research directions?

Given all the above, I recommend acceptance for this paper, but hope the authors can take into account the feedback to further improve the paper.

**Time Spent Reviewing:**

5 hours

---

> ### Author Response · Authors · 2021-08-09
> **Response to Reviewer WNTT**
>
> Thank you for the insightful comments and useful suggestions. We will address your comments below.
>
> > “One limitation of this work is that the approach seems quite similar to prior work which uses successor features for transfer to new tasks (e.g. Barreto et al. 2017, Barreto et al. 2018 Borsa et al. 2018, Ma et al. 2020). [...] I do think the authors should discuss in greater detail how the approach differs from prior work.”
>
> We will include a more clear discussion on distinction to prior work. Summarizing briefly here, the primary distinction is the problem setting for which we leverage SF: to handle nonstationary rewards (and cumulants). This is related to, but distinct from, using SF for related tasks. Consequently, the overall approach is quite different from this prior work--- which primarily considered solving a sequence of tasks---whereas we develop a system that adapts behavior to learn many predictions, leveraging SF in both levels (behavior and prediction).
> We acknowledge that we largely rely on previously developed ideas for SFs to build our system. This can be seen as a strength for the approach, since this means we can leverage more well-tested ideas, but potentially a limitation of the paper, in that there are fewer general purpose algorithmic insights beyond our setting. We would like to note that we do provide some novelty for SF algorithms more generally. The first is the theory highlighting why SF should allow for faster learning. The second is to incorporate interest and reweightings. This is most critical in our setting, where we need to focus learning on sub-areas and handle off-policy samples, but more generally is a useful addition for SF and GPI.
>
>
> > “potential limitation of this work is the reliance upon the choice of policies / tasks to learn [...]  How could one combine this with skill discovery or other related research directions?”
>
> Absolutely the selection of auxiliary tasks matters. In this paper, we choose them in order to (a) yield interesting and difficult-to-learn behaviours, and (b) stress our approach to better understand the strengths and weaknesses of all the algorithmic components. Other choices would result in different outcomes. For example, if all the prediction tasks were constant, each would be learned quickly and the behavior would converge to a random policy. We choose a mix of constant, noisy (replicating a noisy-tv task), and drifting target to highlight that system should be able to: (1) quickly learn easy predictions, (2) learn to ignore predictions when learning stalls, and (3) continually attend to predictions which generate continued learning (drifter). These represent classic, often cited requirements for curious learning systems.
>
> Auxiliary task discovery is still in its infancy, but is synergetic with our approach. For example, Veeriah et al [1] recent meta learning approach represents a first step in GVF discovery. In that work the behavior only focused on external reward. Our approach naturally generates a curriculum because the agent is likely to learn about easier GVFs/aux-tasks first which generate learning progress reward easily, and progress to harder ones once others have been learned or ignored as too difficult. This should combine well with auxiliary task discovery. Naturally discovery is one of the major open questions in AI and although it represents future work, we will certainly expand this discussion in our extra page available for camera ready.
>
>
> [1] Veeriah, V., Hessel, M., Xu, Z., Lewis, R., Rajendran, J., Oh, J., ... & Singh, S. (2019). Discovery of useful questions as auxiliary tasks. arXiv preprint arXiv:1909.04607.

---

> > ### Comment · Reviewer_WNTT · 2021-08-23
> > **Thank you**
> >
> > I thank the authors for their comprehensive response to my comments. I hope the authors will include a more detailed discussion of the related work and limitations of this work in the final draft of the paper, as promised. In conclusion, I think this work would be valuable for the community and recommend it for acceptance!

---

### Official Review · Reviewer_8jZM · 2021-07-16

**Rating:** 8
**Confidence:** 4

**Summary:**

The paper proposes a framework for continually learning multiple generalized value functions, where each value function is learned off-policy and the behaviour policy learns to maximize the learning progress. The paper identifies the non-stationarity of the action value function during the process of maximizing learning progress and proposes an algorithm that separately learns the immediate reward and the successor feature. The experimental results show that the proposed algorithm significantly improves sample efficiency and perfroms well with the use of replay.

**Limitations And Societal Impact:**

The formulation of a GVF question was confusing to me in the first read as it is described as a goal-conditioned value function in Sutton et al. 11'. It seems like the formulation in this paper follows Veeriah et al. 19'. it would be better to clarify this point.

**Main Review:**

### Originality

The idea of using successor feature for MDPs with changing rewards are not new (Barreto et al.18'), but its application in learning auxiliary tasks is novel.

### Quality

The motivation of addressing the non-stationary learning problem in using auxiliary tasks for exploration using successor feature (SF) is very convincing. The paper also shows theoretically that given accurate SF, better convergence can be achieved. The assumption of accurate SF fits the problem setup where rewards are stationary while the dynamics are static. The empirical experiments are also convincing to me.

### Clarity

The paper is well written and easy to follow.

typo: line 277, equation 3 -> equation 2

**Time Spent Reviewing:**

8

---

> ### Author Response · Authors · 2021-08-09
> **Response to Reviewer 8jZM**
>
> Thank you for your review and positive comments!
>
> You are right that the notation is different than the original Horde paper (Sutton et al., 2011). Since the introduction of that work, the notation has been reworked by the authors in subsequent work. The definitive notation is arguably in the recent update to the RL textbook (in 2018), which is precisely the notation we use. The specification is actually equivalent between Horde and the book (namely the same set of GVFs can be specified); however, the new notation is simply cleaner. We will make a note about this notational shift, though we will still cite the Sutton et al. paper for the original introduction of GVFs.

---

> > ### Comment · Reviewer_8jZM · 2021-08-31
> > **Thank you**
> >
> > Thank you for the clarification!

---

### Official Review · Reviewer_NevW · 2021-07-18

**Rating:** 7
**Confidence:** 3

**Summary:**

This paper considers the setting where an RL agent has to learn to make auxiliary predictions, and captures each prediction task as an RL task. In particular, the goal is to maximize the learning progress for each prediction task. Specifically, this is measured by the l1-norm of the change in weights and introduced as an intrinsic reward.

However, this multi-prediction problem is difficult because of multiple sources of non-stationarity, i.e., from the changing rewards (which shrink as the agent makes better predictions) and from the changing agent behavior. To address this non-stationarity, the successor features framework is leveraged. That is, they can use learned successor features to efficiently track the non-stationary rewards and to correct state distribution changes by importance sampling.

**Limitations And Societal Impact:**

Yes. See main review for other limitations.

**Main Review:**

Originality:
- The use of successor features to track and handle non-stationarity is novel. Applying this to efficiently learn multiple auxiliary tasks is also new.

- The introduction also provides a good overview of the related work in the literature, and describes how the ideas in this paper relate to the prior work.

Quality:
- The paper provides a theoretical analysis of the sample efficiency and shows that it is better compared to standard algorithms.

- The algorithm is evaluated in a simple maze environment and in the Mountain Car task, which are pretty simple RL tasks. Learning multiple auxiliary tasks in simpler environments may be more inefficient than solving a main task, when given one. I’m curious how well this approach scales to more complex environments.

- The number of auxiliary tasks studied in the experiments is also quite small (4 in the maze setting and 2 in Mountain Car). It’d be nice to study how a larger number of auxiliary tasks impacts learning.

Clarity:
- Using the change in the weights as the RL reward seems to introduce an unnecessary source of non-stationarity. In the settings studied in the experiments, the auxiliary tasks involve reaching specific goal states. Could the intrinsic reward instead be defined in terms of reaching these goals which would then give stationary reward functions?

- The different parts of Section 4 currently read a bit disconnected and could be better organized. Including a unified algorithm box would be helpful. Also, are the weights of the GVF learners the same weights as in Alg. 2?

- There is a lot of notation and terms that get abbreviated, which gets difficult to keep track of. A table that summarizes all of the notation and abbreviations would be helpful.

Significance:
- Learning a set of auxiliary tasks in the absence of external rewards ideally allows for efficient transfer when an external reward is actually specified, and is hence an important problem setting to study.

- The successor features framework allows for transfer between these auxiliary reward functions, which are assumed to be linear in some features. Is this a limiting assumption when we might want to consider more complex auxiliary tasks that do not share a common feature set?

**Time Spent Reviewing:**

3

---

> ### Author Response · Authors · 2021-08-09
> **Response to Reviewer NevW**
>
> Thank you for your thoughtful review. We address your comments below.
>
> > “Change in the weights as the RL reward seems to introduce an unnecessary source of non-stationarity [...] Could the intrinsic reward instead be defined in terms of reaching these goals which would then give stationary reward functions?”
>
> The goal of the system is to learn value functions, rather than learning how to balance between reaching goals. Rewards based on learning are inherently nonstationary: at the beginning of learning, the agent is unsure about its estimate in a state and changes those estimates more, and later becomes more sure, indicating the behavior need not gather more data there. Rewards based on reaching goals will likely not encourage the agent to get the value function estimates accurate across the space; in fact, it may simply settle on going to the goal or goals that have the highest reward.
> Let us give one more example. The GVF cumulants (specifying the predictions) may not be something we want to maximize. For example, a GVF cumulant might be +1 per-step under the policy of drive forward until bumping into the wall. If this steps-to-wall GVF were accurate, then it could be used as a predictive feature (as in GVF Networks), or as an option for planning. Our approach would stop the agent from bumping into the wall once the prediction is accurate---thus potentially limiting physical damage to the robot. Maximizing the cumulant would not be what we want in this situation.
>
> > “are the weights of the GVF learners the same weights as in Alg. 2?”
>
> Yes.
>
>
> > “A table that summarizes all of the notation and abbreviations would be helpful.”
>
> Good idea.
>
> > “Is this a limiting assumption when we might want to consider more complex auxiliary tasks that do not share a common feature set?”
>
> The transfer here is for nonstationarity in rewards or cumulants, namely across time. Each GVF itself, however, can use its own features and learn its own SF; so it is not a limiting assumption.

---

> > ### Comment · Reviewer_NevW · 2021-08-31
> > **Thanks for the clarifications!**
> >
> > Thanks for the clarifications! I've updated my score after reading the authors' response and other reviews.

---

### Official Review · Reviewer_GBFf · 2021-07-18

**Rating:** 7
**Confidence:** 4

**Summary:**

This work addresses the multi-prediction problem in MDPs, with key focus on how to generate and adapt the behavior to optimize auxiliary predictions in the absence of task rewards.  They show that the rewards for the behavior have inherent non-stationarity. The proposed method approximates successor features (via expected SARSA and Tree Backup) and then uses them to learn value predictions (via regression updates). This specific separation allows for a faster convergence rate as proved by the authors.

**Ethical Concerns:**

None.

**Limitations And Societal Impact:**

The authors do not discuss limitations. I recommended adding this with the conclusion discussing failure modes and future work. As it stands, nothing along these lines have been covered in the manuscript.

I did not see a broader impact statement, although the work is abstract enough to have any direct concerns.


**Main Review:**

**Strengths**:

The work addresses an important problem of learning how to generate behavior such that multiple predictions can be learned, which remains under-explored in the literature.  The approach being fully online adds an additional challenge to tackle under non-stationarity - this is an important contribution. With theoretical and practical considerations, the proposed method seems an effective strategy to learn multiple predictions. This is a very valuable contribution as a lot of the existing literature has explored learning multiple predictions without paying much heed towards learning the behavior policy itself to be able to result in meaningful multiple predictions. While the paper can be significantly improved in writing, the contributions outweigh its flaws.

**Weaknesses**:
* One of the primary limitations of this work is that the cumulants are hand-specified and not being learned or discovered. There is no discussion about this. Any insights would be useful to add. For example how does the work on discovering useful cumulants might tie to the ideas presented in this paper for overcoming user designed cumulant equations?
* The phrasing of the paper title is misleading. The word continual learning is typically used in the context of lifelong learning or never ending learning (also referred to as continual learning). But what the authors really imply here is online learning (e.g “continually interacts with the environment”), which warrants the current title misplaced and not suitable for this work. I would recommend updating it to reflect more clearly what the paper addresses.


**Empirical Analysis**:
The empirical analysis provided support for the claims. The experiments cover a range of tasks. However, I have the following questions:

* In the current pipeline, the value estimates of the prediction have a huge dependence on the accurate learning of the successor features. What happens if the SF is not well learned? Would the method still work?
* How does the change in weights impact the scalability of the algorithm? A potential failure case is that while the weights have changed a lot but nothing meaningful has been learned. How would one interpret this in the context of the proposed approach?
* Besides, it would be interesting and useful to show the failure cases for the proposed ideas in general. Please add additional analysis if any towards this.
* An interesting baseline would be an agent such as Horde where the behaviour policy is not being learned or adapted smartly. Would it be a good idea to add this? Any insights from the authors on this would be appreciated.


**Writing and Presentation**:
* The paper is very well motivated.
* The experiments section is less clear as compared to the other sections. It might be useful to revisit it to ensure better understanding.
* The checklist has many links missing which show as ?
* L359- the above results are obtained by learning online.
* L375 fourth → forth
* I recommend at least one pass of proofread to check the paper for typos and minor writing errors.


**Time Spent Reviewing:**

4

---

> ### Author Response · Authors · 2021-08-09
> **Response to Reviewer GBFf**
>
> Thank you for your thorough review. We will incorporate the writing suggestions, and otherwise address your other comments below.
>
> > “One of the primary limitations of this work is that the cumulants are hand-specified...”
>
> A full system will require both a mechanism to discover predictions (discover cumulants) as well as algorithms to learn those predictions. Both discovery and learning are critical questions, with open questions to be addressed for both in isolation. The focus of this work is on learning multiple GVFs in parallel; as we highlight, there is much more to be done in terms of algorithms and empirical understanding, without also considering the additional complication of discovery.
>
> > “paper title is misleading.”
>
> Here, continual is describing how we are learning auxiliary tasks, that is continually. We can see how this terminology could be misleading. As you likely know, there are many names for continual learning systems, back to early work from Sebastian Thrun in the 90s, including lifelong learning, cumulative learning, never ending learning and continual learning. These have all been used by different communities at different times. In our problem setup, the agent has to continually update its predictions and indeed learning never ends. Nevertheless, today continual learning has certain connotations and, as you say, could be misleading; we will consider alternatives.
>
> > “What happens if the SF is not well learned?”
>
> The weight change of the SF is part of the intrinsic reward, encouraging the behavior to take actions to accurately learn the SF. The approach would still work, even if the SF is not learned perfectly, in the same way that it still works even if the value function approximator cannot learn values perfectly. The parameters for the SF should progressively converge to those possible given the function approximator; once it is done being learned, even if it has some inaccuracies, the behavior will stop focusing on action selection for that SF (and likely focus instead on updating predictions for nonstationary cumulants).
>
>
> > “A potential failure case is that while the weights have changed a lot but nothing meaningful has been learned.”
>
> It is absolutely true that the efficacy of the intrinsic reward relies on the prediction learner being able to modulate learning (what is called being introspective). We leverage this insight from Linke et al. 2021, from the bandit setting, where they show that learning in the whole system (behavior plus predictions) is much more effective if prediction learners modulate down learning when more learning is not possible (e.g., high noise). They do so using stepsize adaptation, as do we. As an example of this in our setting, in the distractor cumulant in the TMaze, once the mean is learned, the noise in the cumulant can induce large updates that are not meaningful. We show that with step size adaptation GVF step sizes decrease until the agent is able to effectively ignore this distraction.
> Nonetheless, our prediction learners will have limitations, and the behavior will be impacted. We demonstrated one such limitation, when features for the cumulant were not well specified (Section 7). With the additional page for the final paper, we can include a longer discussion section on potential failure cases and limitations.
>
>
> > “An interesting baseline would be an agent such as Horde where the behaviour policy is not being learned or adapted smartly.”
>
> The fixed behaviour policy is not a learned policy and provides a baseline as a reasonably smart data collection policy. It would be reasonable to provide a less smart baseline, such as a random policy, to show how much more effective it is to learn a directed behavior. We have tested the random policy previously, and it is much less effective (for example it has 4x higher error than GPI in the last 10%); we will include this baseline in the appendix.

---

> > ### Comment · Reviewer_GBFf · 2021-08-20
> > **Thanks.**
> >
> > Thank you for a detailed response to my concerns and comments. I acknowledge the rebuttal response and most of my questions have been duly addressed. Thanks.

---

### Decision · Program_Chairs · 2021-09-27

**Decision:**

Accept (Poster)

**Comment:**

The reviewers have come to a strong consensus for acceptance of this work. Its strong points include novelty, where the important combination of multi-prediction and non-stationarity is well motivated, as well as pleasing theory and experiments to address the same.

I would ask the authors to carefully consider the suggested writing improvements from GBFf and suggestions for clarity from NevW. I would also like to see additionally comentary about the number of tasks and the potential to discover "useful" tasks/skills in the context of your work (these are not things I expect a solution to today, but something where guidance will help the authors bound to study the topic in the future).